# Identification and characterization of Varicella Zoster Virus circular RNA in lytic infection

Shaomin Yang[1,2,7], Di Cao[1,7], Dabbu Kumar Jaijyan[3,7], Mei Wang[4], Jian Liu[5], Ruth Cruz-cosme[6], Songbin Wu[1], Jiabin Huang[1], Mulan Zeng [3], Xiaolian Liu[4], Wuping Sun[1], Donglin Xiong[1], Qiyi Tang [6,8] ✉, Lizu Xiao[1,8] ✉ & Hua Zhu[3,8] ✉

This study investigates the role of circular RNAs (circRNAs) in the context of Varicella-Zoster Virus (VZV) lytic infection. We employ two sequencing technologies, short-read sequencing and long-read sequencing, following RNase R treatment on VZV-infected neuroblastoma cells to identify and characterize both cellular and viral circRNAs. Our large scanning analysis identifies and subsequent experiments confirm 200 VZV circRNAs. Moreover, we discover numerous VZV latency-associated transcripts (VLTs)-like circRNAs (circVLTs$_{lytic}$), which contain multiple exons and different isoforms within the same back-splicing breakpoint. To understand the functional significance of these circVLTs$_{lytic}$, we utilize the Bacteria Artificial Chromosome system to disrupt the expression of viral circRNAs in genomic DNA location. We reveal that the sequence flanking circVLTs' 5' splice donor plays a pivotal role as a cis-acting element in the formation of circVLTs$_{lytic}$. The circVLTs$_{lytic}$ is dispensable for VZV replication, but the mutation downstream of circVLTs$_{lytic}$ exon 5 leads to increased acyclovir sensitivity in VZV infection models. This suggests that circVLTs$_{lytic}$ may have a role in modulating the sensitivity to antiviral treatment. The findings shed new insight into the regulation of cellular and viral transcription during VZV lytic infection, emphasizing the intricate interplay between circRNAs and viral processes.

Varicella zoster virus (VZV) is a human neurotropic herpesvirus belonging to the alpha herpesvirus subfamily[1]. It exhibits two distinct life cycle phases in humans: lytic (productive) and latent (silent) infection. During a primary infection, VZV replicates in T cells, leading to viremia and dissemination to various organs like skin, causing chickenpox (varicella) in young children, with an annual minimum of 140 million new cases[2,3]. Following the primary infection, VZV establishes a lifelong latent infection in the neurons of the Dorsal root ganglion (DRG) and trigeminal ganglia[4]. However, waning immunity due to aging or immunocompromising can trigger VZV reactivation,

[1]Department of Pain Medicine and Shenzhen Municipal Key Laboratory for Pain Medicine, Huazhong University of Science and Technology Union Shenzhen Hospital, Shenzhen, China. [2]Guangdong Key Laboratory for Biomedical Measurements and Ultrasound Imaging, National-Regional Key Technology Engineering Laboratory for Medical Ultrasound, School of Biomedical Engineering, Shenzhen University Medical School, Shenzhen 518060, China. [3]Department of Microbiology and Molecular Genetics, New Jersey Medical School, Rutgers University, 225 Warren Street, Newark, NJ 070101, USA. [4]Institute of Medical Microbiology, Jinan University, Guangzhou, Guangdong 510632, China. [5]School of Biological Sciences and Biotechnology, Minnan Normal University, Zhangzhou 363000, China. [6]Department of Microbiology, Howard University College of Medicine, 520 W Street NW, Washington, DC 20059, USA. [7]These authors contributed equally: Shaomin Yang, Di Cao, Dabbu Kumar Jaijyan. [8]These authors jointly supervised this work: Qiyi Tang, Lizu Xiao, Hua Zhu. ✉e-mail: qiyi.tang@howard.edu; nsyyjoe@live.cn; hua.zhu@rutgers.edu

causing zoster[5]. Patients suffering from zoster often experience postherpetic neuralgia (PHN)[6], characterized by chronic pain that can significantly impact their quality of life, leading to social withdrawal and, in severe cases, even depression and suicidal tendencies[7].

VZV harbours a large linear double-stranded DNA encoding approximately 71 Open reading frames (ORFs)[8]. During lytic infection, the expressed VZV genes can be categorized into immediate-early (IE), early (E), and late (L) genes[9]. However, during latency, most VZV genes remain silent, except for VZV latency-associated transcripts (VLTs)[10] and VLTs-ORF63 (VLT63) fusion transcripts[11]. Recent advancements in sequencing technologies, such as second-generation short-read sequencing and Oxford Nanopore Technologies (ONT) long-read sequencing, have unveiled a complex structural transcriptome from VZV-infected cells and identified numerous uncanonical ORFs of noncoding RNAs[12]. This transcriptome contains previously undiscovered transcripts, various transcript isoforms, and intriguing splicing events that warrant further investigation[13].

Circular RNA (circRNA) represents a type of non-coding RNA formed through covalently bonding single-stranded RNA via back-splicing in mammalian cells. Its unique circular structure, lacking a 5′ cap and 3′ a poly (A) tail, imparts greater resistance to exoribonuclease, such as RNase R, than linear RNAs[14]. The conventional approach for circRNA analysis requires the identification of back-splice junctions (BSJs) from short-read RNA-seq data treated with RNase R[15]. Recent advancements in sequencing technologies, particularly ONT long-read sequencing, which combines RNase R treatment and rolling circular reverse transcription, provide an advantage in deeply recognizing the full length circRNA[16]. CircRNAs can function as microRNA (miRNA) sponges, regulators of parental gene expression, and translation templates[14,17]. Viral circRNAs have been identified in cells infected with various DNA and RNA viruses, including Epstein-Barr virus (EBV)[18–21], Kaposi Sarcoma herpesvirus (KSHV)[19,22–24], human papillomaviruses (HPVs)[25], human cytomegalovirus (HCMV)[26], Marek's Disease Virus (MDV)[27], and coronaviridae (SARS-CoV-2, SARS-CoV, and MERS-CoV)[28].

In this work, we discover VZV-encoded circRNAs from both VZV-infected cells and HZ patients' lesion fluid. Particularly, we identify a VZV circRNA, named circVLTs$_{lytic}$, derived from VZV latency-associated transcripts (VLTs). Our findings suggest that circVLTs$_{lytic}$ confer acyclovir resistance to VZV, shedding light on a potential mechanism contributing to VZV's ability to evade antiviral treatment.

## Results

### Dynamic landscapes of the host circRNAs during VZV infection

In this study, we investigated the expression kinetics of human circRNAs during VZV lytic infection in human neuroblastoma cells (SH-SY5Y). The study design and analysis were illustrated in Fig. 1A. VZV pOka strain, which contains a green fluorescent protein (GFP) gene between ORF60 and ORF61[29], was used to infect SH-SY5Y cells at an MOI of 0.1 for 24 or 48 h in triplicates, and total RNAs were collected. To validate the VZV infection, we employed fluorescence assay to detect GFP-tagged VZV, showing GFP expression at 24 and 48 hpi (upper panels) and exhibiting cytopathic effect (lower panel) (Supplementary Fig. 1A). Additionally, viral RNA and protein were detected using RT-qPCR (Supplementary Fig. 1B) and western blot assays (Supplementary Fig. 1C). For transcriptome analysis, total RNAs were treated with RNase R and subsequently subjected to deep RNA-seq analysis. The RNA-seq reads were aligned to both the human and the pOka-VZV strain genome using two aligners, Bowtie2 and BWA-MEM[30]. After running the BWA-MEM pipeline, approximately 73,000,000–120,000,000 total reads were obtained from each short reads sequencing samples. Notably, 85 ~ 97% of these reads mapped to the human genome (Supplementary Fig. 1D, E).

Two algorithms, find_circ[31] and CIRI2[32], were utilized to identify cellular circRNAs. The identified cellular circRNAs were subsequently deposited into circBank, a publicly available circRNA database, based on their genomic location. The analysis revealed approximately 15,000–22,000 unique back-splice junctions (BSJs, BSJs counts ≥2) representing putative circRNAs in each sample. Notably, approximately 80% of these circRNAs are cataloged in the circBank database (Fig. 1B and Supplementary Data 1). Interestingly, pooling biological replicates significantly increased genome coverage, resulting in the identification of 48,402 human circRNAs by CIRI2, twice more than with a single sample. To enhance the accuracy of circRNA prediction, we merged BSJs obtained from find_circ with those from CIRI2, considering breakpoints with less than a 10nt difference as corresponding to the same circRNA. Notably, most of the circRNAs identified by CIRI2 were also detected by find_circ, with 35,076 common circRNAs (Fig. 1C). To gain insight into the circRNA landscape, we mapped all the common circRNAs by the 5′ and 3′ breakpoints of the BSJs to their respective chromosomal locations and estimated the back-splicing frequency by counting BSJ-spanning reads (Fig. 1D). Results demonstrated that most cellular circRNAs were produced at a basal level that was not affected by VZV infection.

Subsequently, differential expression analysis of cellular circRNAs (DE-circRNAs) was performed using DESeq2[33]. Our results revealed 532 upregulated DE-circRNAs and 253 downregulated DE-circRNAs in SH-SY5Y cells infected with VZV for 24 h (Fig. 1E), and 771 of upregulated and 318 downregulated DE-circRNAs in the cells infected with VZV for 48 h (Fig. 1F). To delve deeper into the characteristics of the parental genes associated with VZV regulated DE-circRNA, we performed Gene Ontology (GO) and Kyoto Encyclopedia of Genes and Genomes (KEGG) pathway analyses. The analysis for Biological processes indicated an association with "regulation of cell growth" and "chromosome segregation" for the parental genes of VZV-regulated DE-circRNAs (Fig. 1G). The GO Cellular Component analysis revealed the enrichment in the "spindle" (Fig. 1H). Molecular function analysis identified enrichments in "GTPase regulator activity" and "nucleoside-triphosphatase regulator activity" (Fig. 1I). In addition, the KEGG pathways analysis indicated enrichment of the "Axon guidance" and the "Thyroid hormone signaling pathway" pathway (Fig. 1J). A heatmap plot showed that the significant DE-circRNAs were regulated after VZV infection (Fig. 1K). Furthermore, we selected 6 significant VZV-regulated cellular DE-circRNA for verification of their junction sites and expression using Sanger-sequencing and qPCR with divergent primers (Fig. 1L, M). Taken together, the features of cellular circRNAs during VZV infection were characterized.

### Computational identification and characterization of VZV circRNAs

Next, we aimed to identify and characterize VZV-encoded circRNAs. To comprehensively identify VZV-encoded circRNAs, we applied two sequencing technologies: BGISEQ short-read and ONT long-read sequencing. The ONT long-read sequencing combined with RNase R treatment and rolling circular reverse transcription (Fig. 2A). Three reference annotation-independent algorithms, find_circ, vircircRNA[25] and CIRI2 with downstream analysis tool CIRI-full for de novo reconstruction of full-length circRNAs[34], were employed to analyze short-read data, and CIRI-long was used to analyze the long-read data[16]. The identified VZV circRNAs were cataloged in ViruscircBase[35] based on their genomic locations, the only publicly accessible viral circRNA databases (Supplementary Fig. 2A). CIRI2 (Supplementary Fig. 2B and Supplementary Fig. 2F), find_circ (Supplementary Fig. 2C and Supplementary Fig. 2G), and vircircRNA (Supplementary Fig. 2D and Supplementary Fig. 2H) identified about an average of 16, 49 and 91 VZV circRNAs in each biological replicate, respectively. As expected, the count of viral circRNAs increased with the duration of VZV infection. Recognizing the benefits of improved genome coverage through pooling biological replicates (Fig. 1B), we combined all short-read sequencing replicates with VZV infection as one sample for VZV circRNA detection. A total of 2,806, 106, and 305 unique short-read BSJs

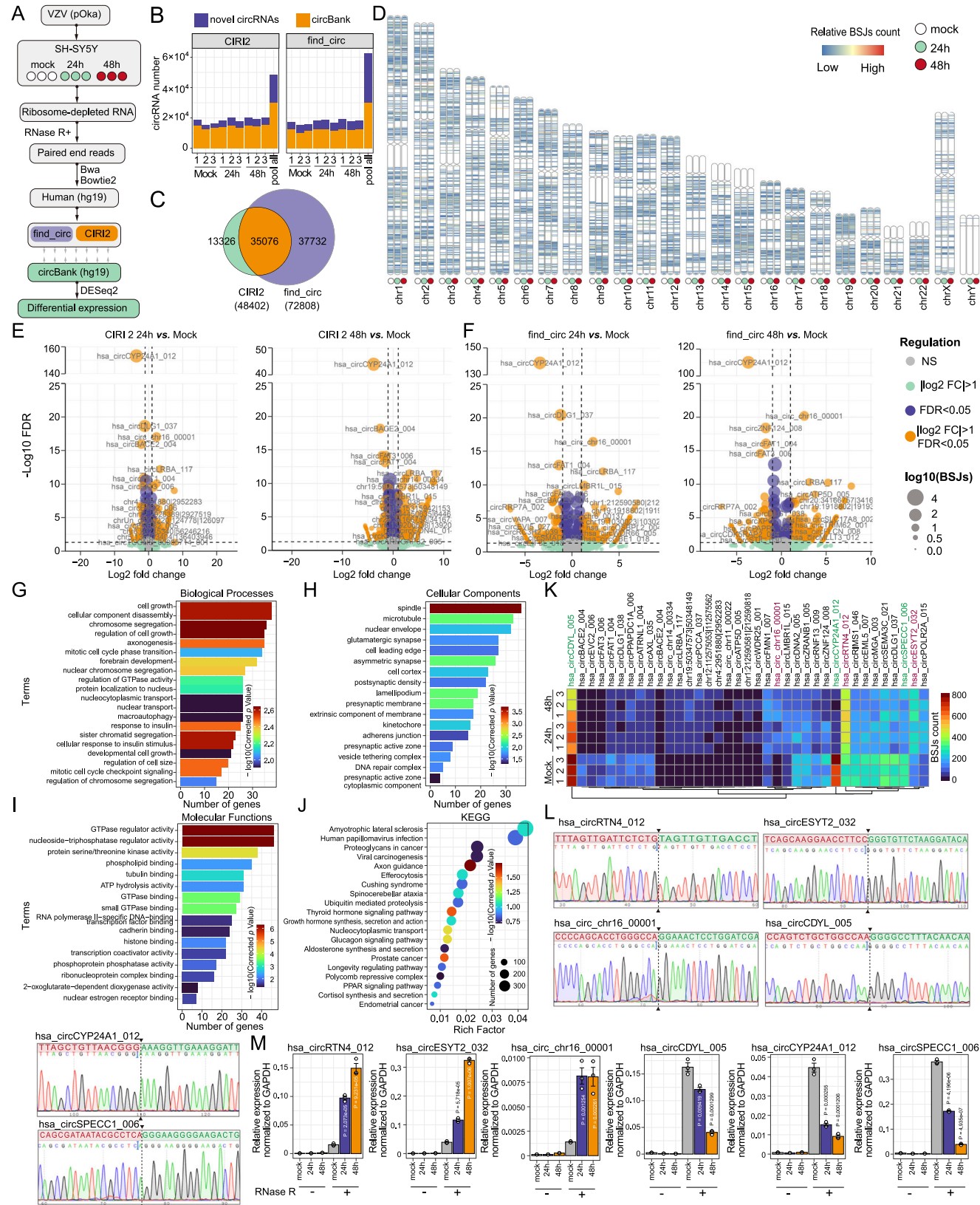

representing VZV-encoded circRNAs were predicted by find_circ, CIRI2 and vircircRNA, respectively (Supplementary Data 2). Additionally, 106 full-length VZV circRNAs were reconstructed using CIRI-full (Supplementary Data 2). Significantly, CIRI-long identified 1358 full-length VZV-encoded circRNA from ONT long-read sequencing data (Supplementary Fig. 2E and Supplementary Data 2). Unfortunately, only 78

VZV circRNAs are available in ViruscircBase, based on mRNA-seq data without RNaseR enrichment[35]. Consequently, small amounts of VZV circRNAs identified in this study were included in ViruscircBase (Supplementary Fig. 2B–E).

To assess the performance of the two sequencing technologies for full-length cellular and VZV circRNA detection and reconstruction,

**Fig. 1 | The landscape of cellular circRNAs during VZV infection. A** Workflow of circRNAs identification. SH-SY5Y cells were infected with VZV pOka strain for 24 and 48 h, with three replicates per group. Total RNA was treated with RNase Rand subjected to deep sequencing. RNA-seq reads were aligned to the human genome, and circRNAs were identified using find_circ and CIRI2. CircRNAs were annotated by a publicly available database, circBank. Finally, differential expressions of circRNAs were screened by DESeq2. **B** Statistics of human circRNAs in each group or pooled biological replicates. **C** Venn diagram presenting unique or shared human circRNAs, identified by CIRI2 and find_circ. **D** Chromosomal distribution of cellular circRNAs. **E, F** Volcano plot showing differentially expressed human circRNAs at 24 (**E**) and 48 h (**F**) post-VZV infection. Log2 (fold change) is plotted as the abscissa, and log10 (q value) is plotted as the ordinate. The mean of BSJ read is plotted as point size. **G–I** Gene Ontology (GO) enrichment annotation of human DE circRNAs from host source genes, including Biological Processes (**G**), Cellular Components (**H**), and Molecular Functions (**I**). **J** Kyoto Encyclopaedia of Genes and Genomes (KEGG) pathways of human DE circRNAs from host source genes. **K** Heatmap of top 50 DE-circRNAs from each sample. **L** Sanger sequencing results of the BSJ sequences of significantly DE-circRNAs. **M** qPCR was used to detect the expression changes of DE-circRNAs. $N = 3$ independent experiments were performed. Statistical comparisons were conducted with a two-sided unpaired t-test. The P value of VZV infection 24 or 48 h *vs.* mock group was shown. Data are presented as mean ± S.E.M. Source data are provided as a Source Data file.

we visualized the correlation between the circRNA length and expression levels in a density heat map (Fig. 2B–E). Most human and VZV circRNAs are shorter than 500 nt, with longer circRNAs exhibiting lower expression levels. As expected, short-read sequencing, with its high-throughput advantage, reconstructed 42,563 full-length human circRNAs, as well as additional partially assembled circRNAs using circfull. In contrast, long-read sequencing identified 12,407 cellular circRNAs. However, for VZV circRNA identification, short-read sequencing identified 71 full-length VZV circRNAs, while long-read sequencing identified 1358 full-length VZV circRNAs, exhibiting a similar length-expression distribution trend to cellular circRNAs. These data suggest the efficiency and reliability of rolling circular long-read sequencing for viral circRNA detection.

To evaluate the tolerance of the VZV transcriptome to RNase R digestion, we analysed the genome coverage of VZV in ribosome-depleted RNA samples with or without RNase R treatment. Short-read sequencing showed approximately 2 ~ 3%, 6 ~ 7%, and 12 ~ 13% of the reads mapping to the VZV genome 24, 48, or 72 h post-infection (Supplementary Fig. 1D, E, and Supplementary Fig. 3A–D). Long-read sequencing detected 15.11% of 3,499,429 total reads mapping to the VZV genome (Supplementary Fig. 1D, E, and Supplementary Fig. 3E). Interestingly, specific regions, including 10000–15000 bp (ORF9-ORF9A) and 110,000–135,000 bp (ORF61-ORF71), exhibited resistance to RNase R treatment (Supplementary Fig. 3), indicative of circRNA formation. Mapping the BSJ positions and expression levels to a scatter plot, predicted by CIRI2 (Fig. 2F), find_circ (Fig. 2G), vircircRNA (Fig. 2H) and CIRI-long (Fig. 2I) revealed similar distribution patterns of VZV circRNAs with most being local BSJs (spanning distance less than 2 Kb) and abundant in hotspot regions, particularly 110,000–135,000 bp (ORF61-ORF71). Among these VZV circRNAs, 3 VZV circRNAs were found to be common to all four algorithms (Fig. 2J), indicating significantly different results of these algorithms.

Comparing the characteristics of VZV and human circRNAs in these two sequencing technologies, including contained exons, strand preferences, and length, reveals several key findings. About 73.2% of VZV circRNAs contained multiple circular exons, whereas 94.6% of cellular circRNAs exhibited this feature (Fig. 2K). In addition, statistical analysis indicated that VZV circRNAs, similar to human circRNAs, exhibited approximately 50% strand preference (Fig. 2L). Furthermore it was revealed that the average length of VZV circRNAs was shorter than that of cellular circRNAs (Fig. 2M).

Given the reported role of circRNAs in regulating the expression of their parental genes[36,37], and in light of our previously constructed genome-wide mutagenesis library for functionally annotating the 71 VZV ORFs[8], we matched annotated full-length circRNAs to specific locations in VZV genome reference with each ORF. The growth properties of its corresponding virus gene deletion are shown in a green or yellow color in Fig. 3A. This analysis led to the identification of multiple isoforms of circRNAs (Fig. 3B, C), such as circVLTs$_{lytic}$ (114003 | 112034), located in ORF61 with multiple exons and presenting multiple isoforms of circRNAs. This circVLTs$_{lytic}$ exhibited homology

with VZV latency-associated transcripts (VLTs)[10] that exon 2 of VLTs$_{lytic}$ was spliced with exon 5 of VLTs$_{lytic}$ (Fig. 3D and Supplementary Fig. 4). Taken together, we provide a glimpse into the landscape of VZV-encoded circRNA.

## Experimental confirmation of the viral circRNAs from VZV-infected cells

To experimentally confirm the presence of VZV circRNAs in VZV-infected cells, total RNA was extracted from SH-SY5Y cells infected with VZV pOka strains for 48 h. Divergent primers were designed to amplify the targeted BSJs of the most abundant VZV-encoded circRNAs (Supplementary Fig. 5A, and Supplementary Data 3). Additional two divergent primer sets were used to obtain VZV circRNAs in hotspot regions, including 10,000–15,000 bp (ORF8-ORF9) and 110,000–135,000 bp (ORF61-ORF71) (Supplementary Fig. 5B). To validate the BSJ amplicons and ensure that they were from genuine circRNAs, gel-purification was performed on candidate BSJ amplicons based on molecular weight. Subsequently, these amplicons were subcloned, and Sanger-sequenced from at least 8 colonies for each candidate. We randomly selected 8 primer sets, targeting VZV circRNAs identified from short-reads and long-reads sequencing data to perform a small sample test (Supplementary Fig. 5C, D, Supplementary Fig. 5F and Supplementary Data 3). We identified 8 BSJs from 8 primer sets (Supplementary Fig. 5H), with BSJ#4 corresponding to circVLTs$_{lytic}$. Subsequently, employing this pipeline, we conducted a comprehensive large-scale genome-wide scanning experiment (Fig. 4A), leading to the identification of a total of 200 VZV circRNAs from 235 clones (Supplementary Data 3).

To assess the resistance of VZV circRNAs to RNase R digestion, we subjected total RNA extracted from pOka-infected SH-SY5Y cells to RNase R treatment. Agarose gel electrophoresis revealed that ribosomal RNAs and cellular or viral linear RNAs derived from GAPDH or ORF28 were mostly degraded after 20 min of RNase R treatment (left side of Fig. 4B). In contrast, the well-characterized cellular circRNA (circHIPK3), and viral circRNAs, including BSJ#1 (42280 | 41838), BSJ#2 (77973 | 75876), BSJ#3 (86381 | 86079), VZV circVLTs$_{lytic}$ [BSJ#4 (114003 | 112034)], BSJ#6 (100697 | 100292), and BSJ#7 (113995 | 112096) remained largely unaffected or even enriched. However, BSJs of BSJ#5 (85940 | 85375) and BSJ#8 (120396 | 119995) were more susceptible to RNase R digestion (Fig. 4B, Supplementary Fig. 5E and Supplementary Fig. 5G). These results suggest that most VZV BSJ-containing RNAs are resistant to RNase R digestion. Our bioinformatics analysis revealed that circVLTs$_{lytic}$ encompasses multiple exons and exhibits various isoforms (Fig. 3B, C). To target exon 1 of circVLTs$_{lytic}$, we designed divergent primers (Fig. 4C). Subsequently, agarose gel electrophoresis displayed multiple bands (Fig. 4D). Additionally, Sanger sequencing results confirmed the presence of multiple exons in circVLTs$_{lytic}$ (Fig. 4E).

To avoid false circRNA predictions due to template switching during cDNA synthesis, we employed a Fluorescence in situ hybridization (FISH) method with high fidelity for the detection of HCMV and

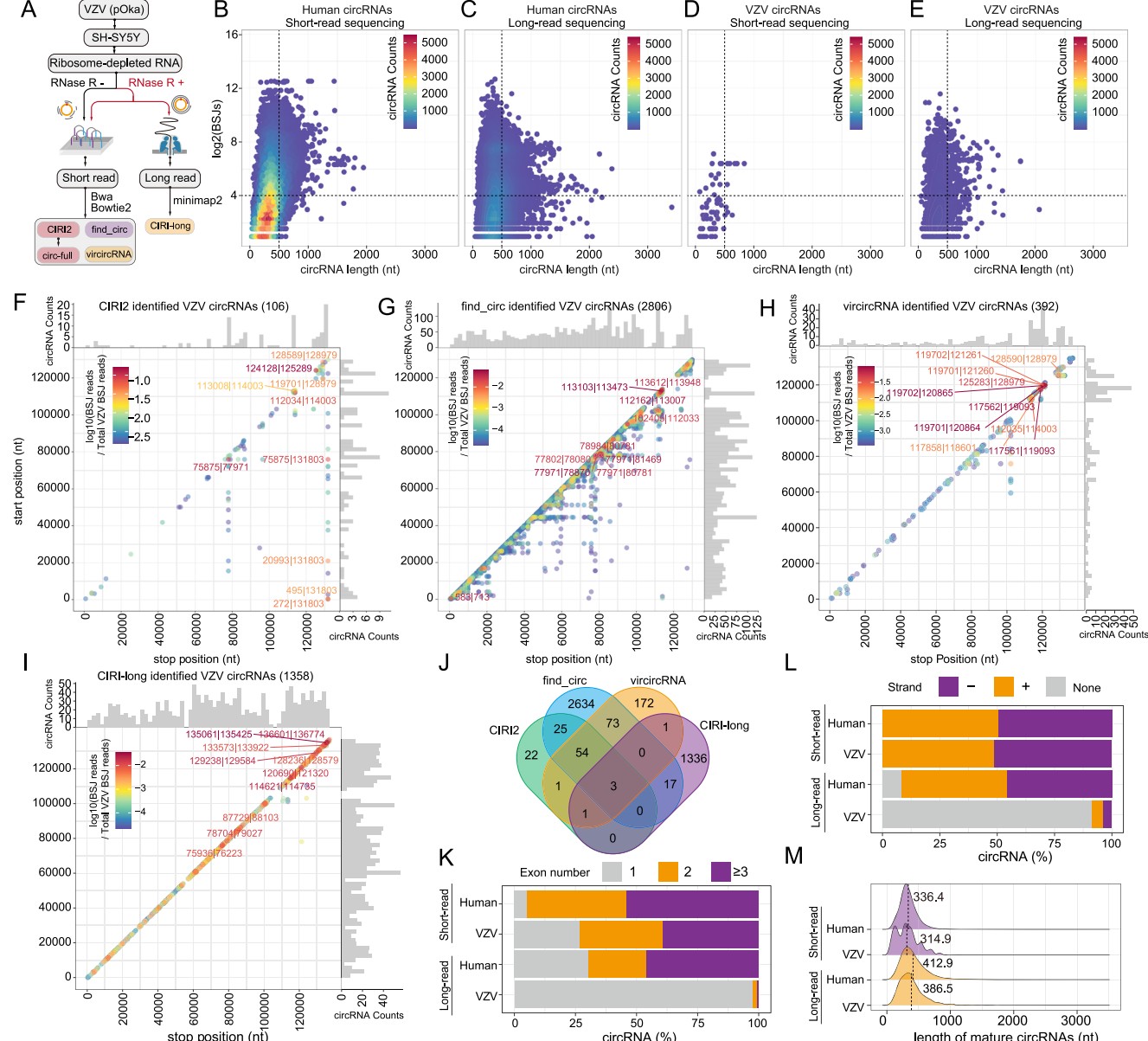

**Fig. 2 | Identification of VZV circRNAs and their characterizations. A** Workflow of VZV circRNAs identification. BGISEQ short-read sequencing data were pooled and analyzed using CIRI2, find_circ, and vircircRNA. ONT long-read sequencing data with rolling circular reverse transcription were analyzed with CIRI-long.
**B–E** Density distribution of length and expression levels of human circRNA identified by short read sequencing (**B**) and long-read sequencing (**C**), and VZV circRNA identified by short read sequencing (**D**) and long-read sequencing (**E**).
**F–I** Frequency of VZV circularization events predicted by CIRI2 (**F**), find_circ (**G**), vircircRNA (**H**) and CIRI-long (**I**). Counts of BSJ-spanning reads (starting from a

coordinate in the X axis and ending in a coordinate in the y axis) were indicated by color. CircRNA counts were aggregated into 20000 bp bins for both axes. Distribution of circRNA counts number was shown as histograms on the x and y axis.
**J** Venn diagram presenting the number of unique or shared VZV circRNAs identified by CIRI2, find_circ, vircircRNA and CIRI-long. **K** Statistics of the exon numbers of circRNAs in humans and VZV. **L** Strand preferences of the circRNAs in human and VZV. **M** Length distribution of circRNAs in humans and VZV. The average lengths are indicated by dashed lines. Source data are provided as a Source Data file.

SARS-CoV-2 circular RNAs[28,38]. The FISH method utilized adjacent probes hybridized to the BSJs, generating a signal only when the circular RNA is present, therefore, this method is specific in detecting circular RNAs (Fig. 4F). As shown in Fig. 4G, VZV circVLTs$_{lytic}$ distribute in the nuclei and cytoplasm of the infected cells. Uninfected cells or single probes against the acceptor sequence showed no ampFISH signal. Therefore, using this method, we demonstrated the presence of VZV circVLTs$_{lytic}$ in both the nuclei and cytoplasm of VZV-infected cells. Taken together, our experimental evidence, including RT-PCR, RNase R sensitivity assay, and the FISH method, confirms the existence of viral circRNAs in VZV-infected cells.

## Identification of VZV circRNAs from tissues of Herpes Zoster patient and clinical strain

To explore the presence of viral circRNAs in Herpes Zoster (HZ) patients, we enrolled six individuals diagnosed with VZV reactivation-induced HZ, characterized by VZV reactivation-caused skin manifestations (Fig. 5A). Information on patients' age, gender, and pain intensity score (measured using a visual analog scale, VAS)[39] was collected and summarized in Fig. 5B. Subsequently, we isolated total RNAs from the blister fluids of the HZ patients and detected viral circRNAs. By employing six primer sets (Fig. 5C, D), we successfully detected 17 VZV circRNAs in the blister fluid samples (Supplementary Data 3),

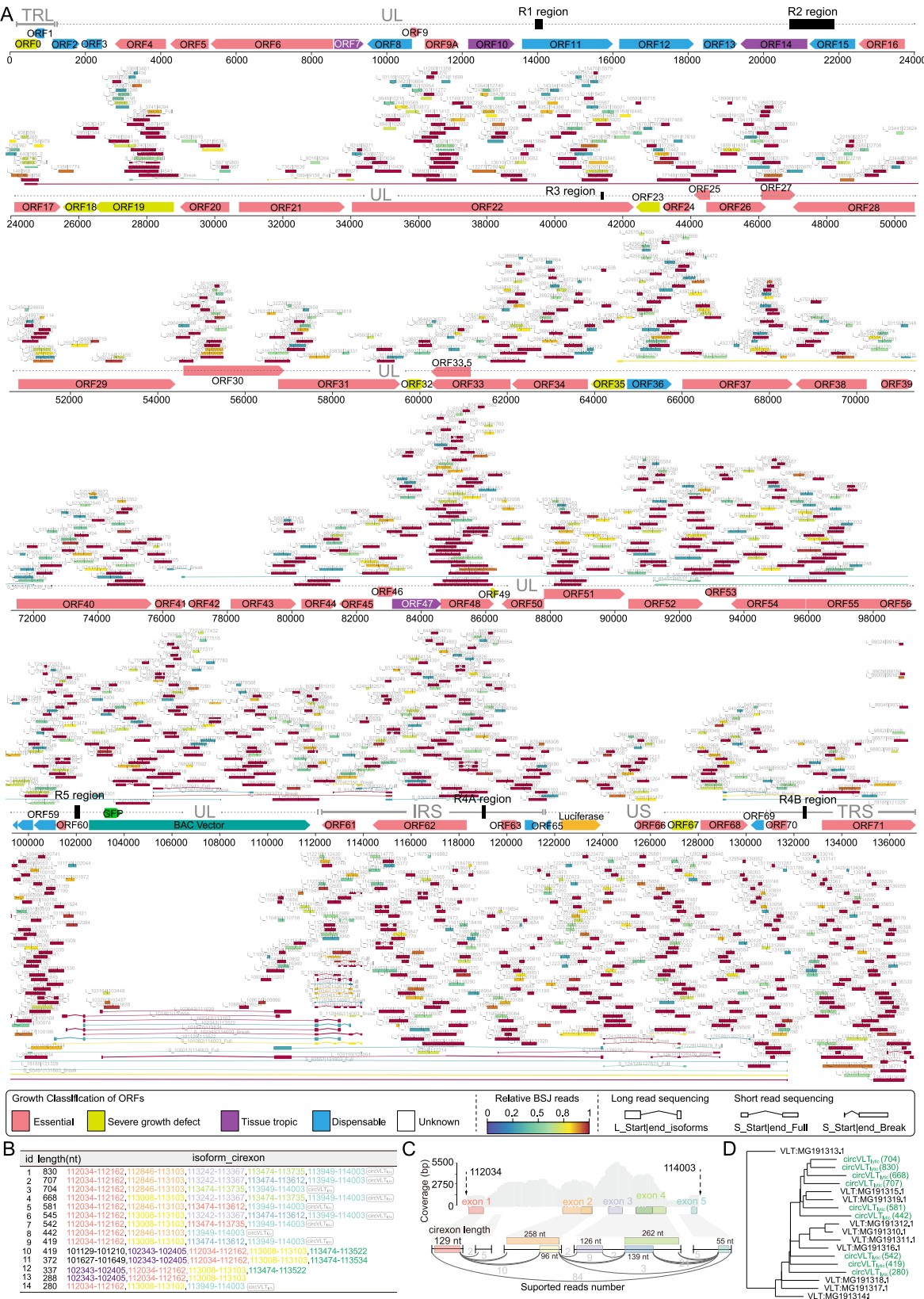

confirming the presence of these viral circRNAs in HZ patients (Fig. 5C, D). Furthermore, we isolated three clinical VZV strains from the rash tissue of HZ patients. Inverse RT-PCR and Sanger sequenced results indicated presence of VZV-encoded circRNAs in these clinical strains (Fig. 5E). Among the identified circRNAs, VZV circVLTs$_{lytic}$ were found in both blister fluids of HZ patients and clinical VZV strains (Fig. 5D).

## The DNA flanking circVLTs's 5' splice donor is a cis-acting element for the formation of VZV circVLTs$_{lytic}$

To understand the biogenesis and function of VZV circVLTs$_{lytic}$ (114003|112034), we employed the Bacteria Artificial Chromosome (BAC) system to investigate the role of the DNA flanking the 5' splice donor of circVLTs$_{lytic}$ in the formation of these circRNAs. The splicing

**Fig. 3 | Full length of VZV circRNAs and circVLTs_lytic. A** Full length of circRNAs reconstruction by BGISEQ short-read sequencing and nanopore ONT long-read sequencing and VZV genome organization of pOka strain with GFP and luciferase gene. The ORFs, unique and the repeat regions (TRL, IRL, UL, IRS, US, TRS) and multiple short reiterations regions (R1, R2, R3, R4, R5) in loci on the VZV genome map were included[69]. The ORFs were color-coded according to the growth properties of their corresponding virus gene-deletion mutants[8]. The relative expression levels of circRNAs are indicated by the colors at the bottom. Reconstruction of full-length circRNAs and partially assembled circRNAs using circ-full were indicated by "Full" and "Break" respectively. **B** The ordinate of each isoform exon of multiple isoforms circRNAs. **C** Isoform-level quantification of VZV circVLTs_lytic. The genome coverage (top) and length (middle), the quantified supported splicing reads number of each cirexon (bottom). **D** A phylogenetic tree of VLTs[10] and VZV circVLTs_lytic. The multiple sequence alignment was shown in Supplementary Fig. 4. Source data are provided as a Source Data file.

breakpoint in genomic DNA was eliminated to disrupt the expression of VZV circVLTs_lytic. Sequence analyses revealed that exon 1 of VZV circVLTs_lytic (112034–112162) overlaps with a uncanonical ORF[12], and exons 2, 3, and 4 (113008–113612) of VZV circVLTs_lytic overlap with ORF61 (112262–113665), with an exception of exon 5 (113949–114003), which has no overlapping with other ORFs (Fig. 6A, B), indicating that the 5' splice donor (114003) is crucial for circVLTs_lytic expression. To explore the significance of this cis-acting element, specific mutations were introduced around the 5' splice donor region. Two different mutations were created: pOka-M1, where 20 bp sequences downstream of the 5' splice donor (114004–114024) were replaced with random sequences, and pOka-M2, where 20 bp sequences upstream of the 5' splice donor (113983–114003) were similarly replaced (Fig. 6C and Supplementary Fig. 6A, B). Both mutations resulted in significant decrease in the expression of VZV circVLTs_lytic (Supplementary Fig. 6B). Further examination using Sanger sequencing and TA cloning revealed that pOka-M2 induced more mis-splicing events than pOka-M1, although most of the circVLTs_lytic of pOka-M2 contained VLTs_lytic sequences (Fig. 6D, E, Supplementary data 4).

To assess the effects of the mutations of circVLTs_lytic on VZV infection, we performed RNA-seq to quantify the expression of cellular and viral genes (Supplementary Fig. 6D, E). A small number of cellular differentially expressed genes (DEGs) (Fig. 6F, G) were identified, and after comparing to pOka-WT (Supplementary Fig. 6F), our results demonstrated that the overall impact on host cells was minor. We quantified the genome coverage of VZV RNA, showing that the RNA transcription was inhibited slightly by mutation of circVLTs_lytic exon 5 (Supplementary Fig. 6F, G). Interestingly, the coverage of VZV circVLTs_lytic exon 1 (112034–112162), far from the mutation position, was reduced in both pOka-M1 and pOka-M2 as compared to that of exon 5 (113949–114003) that was only decreased in pOka-M1, suggesting that exon 5 interacted with exon 1 (Fig. 6H). This observation was confirmed by quantitative real-time PCR (Fig. 6H), suggesting that the DNA sequence flanking the 5' splice donor plays a critical role in the formation of VZV circVLTs_lytic. To evaluate the impact of circVLTs_lytic' mutations on VZV RNA splicing, we performed ViReMa analysis of RNA splicing as we did in our previous studies[26]. The results indicated that the majority of RNA splicing in pOka-M1 and pOka-M2 mirrored that of pOka-WT (Supplementary Fig. 6H–J).

In conclusion, the study highlights the importance of the DNA sequence surrounding the 5' splice donor of circVLTs_lytic as a cis-acting element essential for their biogenesis and expression in VZV-infected cells.

### Mutation of circVLTs_lytic exon 5 makes VZV more sensitive to acyclovir

To investigate the biological functions of viral circRNAs in VZV replication and pathogenesis, we conducted a series of experiments using VZV pOka-WT (wild-type), pOka-M1, and pOka-M2, in which exon 5 of VZV circVLTs_lytic was mutated. VZV pOka genome was inserted with a firefly luciferase reporter between ORF65 and ORF66 and a green fluorescent protein gene between ORF60 and ORF61 (Fig. 2A)[29] so that the growth of VZV can be monitored by fluorescent microscopy or bioluminescence. The growth curves of VZV pOka-WT, pOka-M1, and pOka-M2 were analysed in ARPE-19 cells and it was observed that

mutation of VZV circVLTs_lytic exon 5 did not impair VZV growth, as there were no significant differences in viral replication among the three groups, except on day 3 (Fig. 7A). Next, we assessed the viral sensitivities to antivirals, including interferons (IFNs), acyclovir (ACV), and rhodomyrtone (RDT, a novel anti-VZV agent developed in our laboratory)[40]. The ARPE-19 cells were infected with pOka-WT, pOka-M1, and pOka-M2 for 24 h and then treated with IFN-β, acyclovir, or RDT for 48 h. As shown in Fig. 7B, the same bioluminescence signal of each group was detected before antiviral treatment. The results showed that pOka-M1 displayed increased sensitivity to acyclovir treatment compared to the other two strains (Fig. 7C). To further elucidate the function of circVLT, we overexpressed both the circVLT 419 nt and circVLT 542 nt isoforms in SH-SY5Y cells using a lentiviral circular RNA overexpression vector, followed by infection with pOka-M1. Our results indicate that overexpression of either circVLT isoform enhanced the replication of pOka-M1 and alleviated the antiviral effect of ACV (Supplementary Fig. 7A–D). To further investigate the effects of circVLTs_lytic mutation on viral sensitivity to acyclovir, human skin and DRG were infected for 24 h and then treated with different concentrations of acyclovir. The growth curves of pOka-WT, pOka-M1, and pOka-M2 were monitored by detecting bioluminescence signals every two days. The mutation downstream of circVLTs' 5' splice donor consistently demonstrated sensitivity to acyclovir in these VZV infection models, including in vitro cultures of human skin (Fig. 7D and Supplementary Fig. 7E) and human DRG (Fig. 7E and Supplementary Fig. 7F). In addition, human skin tissues were subcutaneously engrafted into CB-17 SCID/beige mice and the mice were infected with pOka-WT, pOka-M1, and pOka-M2, followed by treatment with acyclovir at different doses. The results in the xenografts SCID mice model further confirmed that the mutation downstream of circVLTs' 5' splice donor also displayed sensitivity to acyclovir in vivo (Fig. 7F, G). These findings collectively suggest that circVLTs_lytic are conditionally required for VZV replication, and their presence or absence can impact the sensitivity of VZV to antiviral treatments, particularly acyclovir.

The VZV thymidine kinase (TK) gene, ORF36, is involved in the antiviral activity of acyclovir that inhibits the synthesis of viral DNA. In addition, VZV expresses VLTs as a factor that antisense inhibits the expression of immediate early gene (IE), ORF61[10]. VLTs also splice with ORF63 (IE gene), forming VLT63 (Fig. 7H)[11]. Thereby, we detected the expression of ORF36, ORF61, VLTs_lytic, VLT63_lytic (Fig. 7H), two viral DNA polymerase catalytic subunit genes (ORF16, ORF28, early gene, E) and a late (L) gene, ORF62, using qPCR. The results showed that the mutation of circVLTs_lytic do not impact the expression of ORF63 and VLT63. However, with ACV treatment, mutation of circVLTs_lytic significantly inhibited the expression of ORF16, ORF28, ORF36 and VLTs_lytic (Fig. 7I) in both pOka-M1 and pOka-M2. Interestingly, ACV treatment disturbed the expression of ORF61 and ORF62 in pOka-M1 but not pOka-M2. These findings suggest that circVLTs_lytic were beneficial to VZV gene expression in responding to antiviral treatment.

## Discussion
CircRNAs have been identified from multiple types of cell lines, tissues, and organs in humans and demonstrated to be associated with different diseases. Recent research has revealed a repertoire of circRNAs derived from viruses, including DNA viruses (EBV, KSHV, HPV, HCMV,

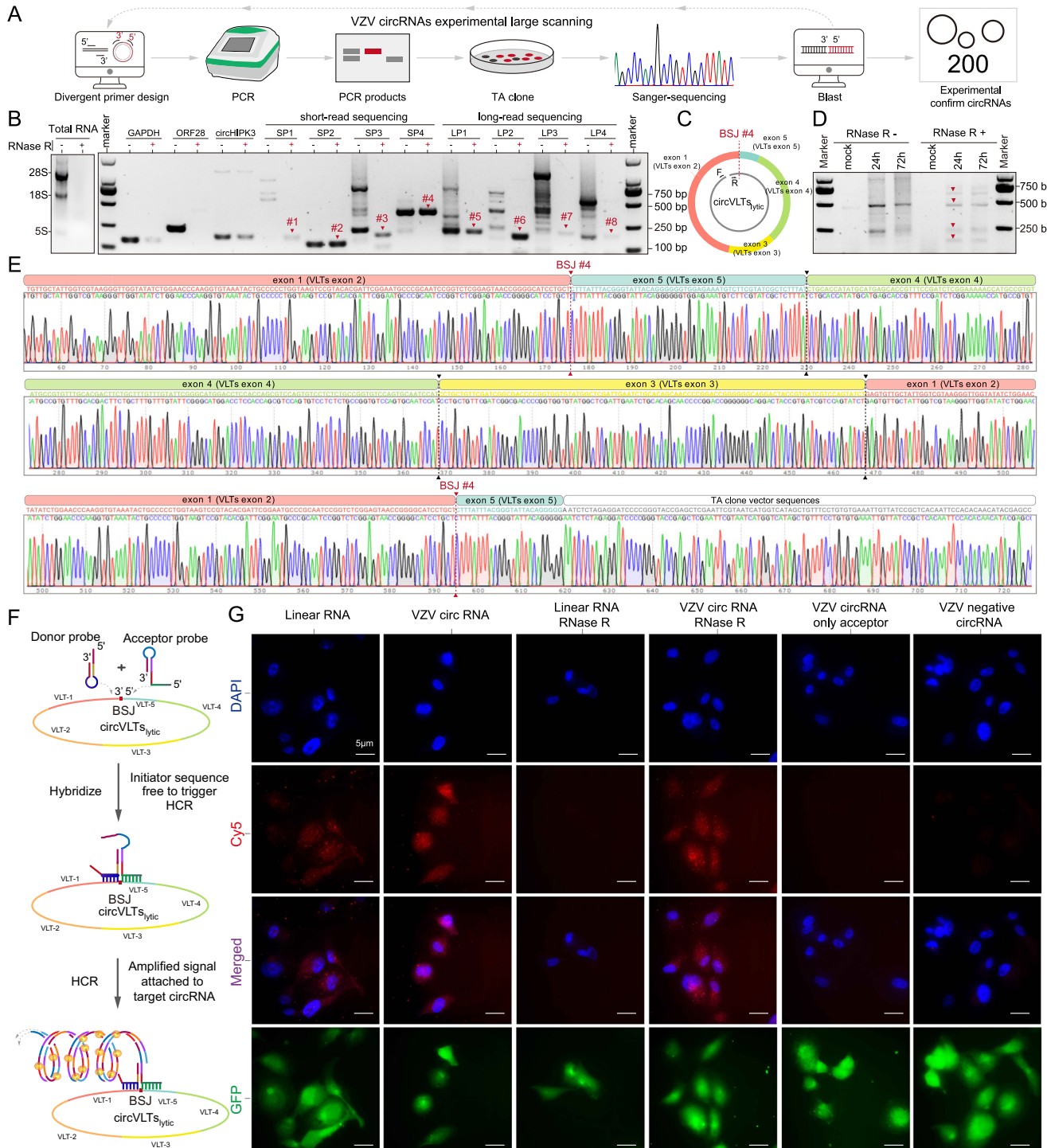

**Fig. 4 | Experimental identification of VZV circRNAs. A** Workflow of circRNAs experimental large scanning. **B** Relative resistance ability to RNase R of linear RNA and circRNAs. The value was normalized to the corresponding sample without RNase R treatment. The left panel shows an agarose gel analysis of total RNA with and without RNase R treatment, and the right panel shows the products of RT-PCR on RNA treated with and without RNase R. Human linear RNA, GAPDH, and VZV linear RNA, ORF28, Human circRNA, circHIPK3, and VZV circRNAs. **C** Schematic diagram showing divergent primer sets designed to amplify the full-length of circVLTs_lytic. **D** Inverse PCR results with divergent primer sets to target vcDNA, Sanger sequencing result shown in (**E**). **F** A schematic illustrating ampFISH. When two hairpin-shaped probes bind at adjacent locations on a target RNA,

conformational reorganization occurs in one of them, which initiates a hybridization chain reaction (HCR) that deposits fluorescence at the target RNA molecule. The target sequences of the two probes are present next to each other in the circular RNA but not in the linear RNAs. **G** Representative VZV-infected ARPE-19 cells imaged after ampFISH either against linear RNA or VZV circVLTs_lytic. Cells harboring VZV are GFP-positive. AmpFISH images are maximal intensity merges of z-stacks acquired in the Cy5 channel and presented under a common contrast level. Scale bar: 5 μm. For (**B**, **D**, **G**), PCR experiments were conducted three times, and representative agarose gel electrophoresis images are presented. Scale bar = 5 μm. Source data are provided as a Source Data file.

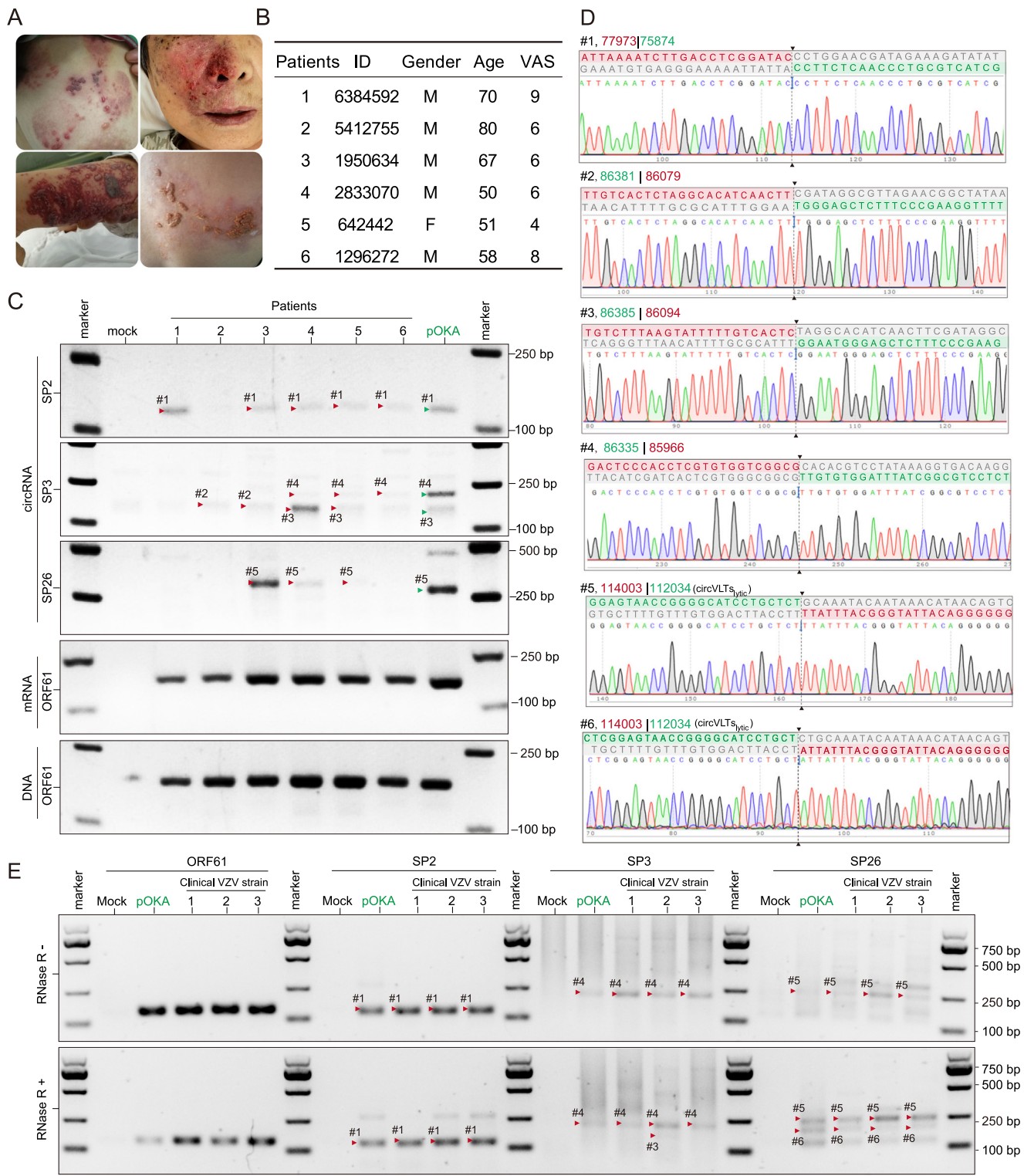

**Fig. 5 | Identification of VZV circular RNA from HZ patients. A** The features of vesicles and pustules in HZ patients. **B** The general information of HZ patients. Pain intensity was assessed using the visual analog scale (VAS), which measures pain intensity on a scale of 0 to 10, 0 indicating no pain and 10 indicating the highest pain. **C** Inverse RT-PCR results with divergent primer sets targeted VZV circRNAs in HZ patients' blister fluids. Representative Sanger sequencing results were shown in

(**D**). BSJ breakpoints are indicated by dashed lines. Donor (green), acceptor (red) sequences, and downstream/upstream sequences (grey) flanking the junction were aligned with the BSJ sequence. **E** Inverse RT-PCR result of the VZV clinical strain. For panels (**C**, **E**), PCR experiments were conducted three times, and representative agarose gel electrophoresis images are presented. Source data are provided as a Source Data file.

and MDV)[18–21,25–27] and RNA viruses (SARS-CoV-2, SARS-CoV, MERS-CoV, and Murine Hepatitis Virus, MHV)[28,41]. Our presented studies are unique because we applied deep circRNA sequencing to landscape circRNAs in VZV-infected SH-SY5Y cells and tissues of herpes zoster

patients that were then experimentally confirmed. Three circRNA identification algorithms were utilized to analyse the experimental data and 200 VZV-encoded circRNAs were identified. Upon comparing the different algorithms, CIRI2 was found to be the most suitable for

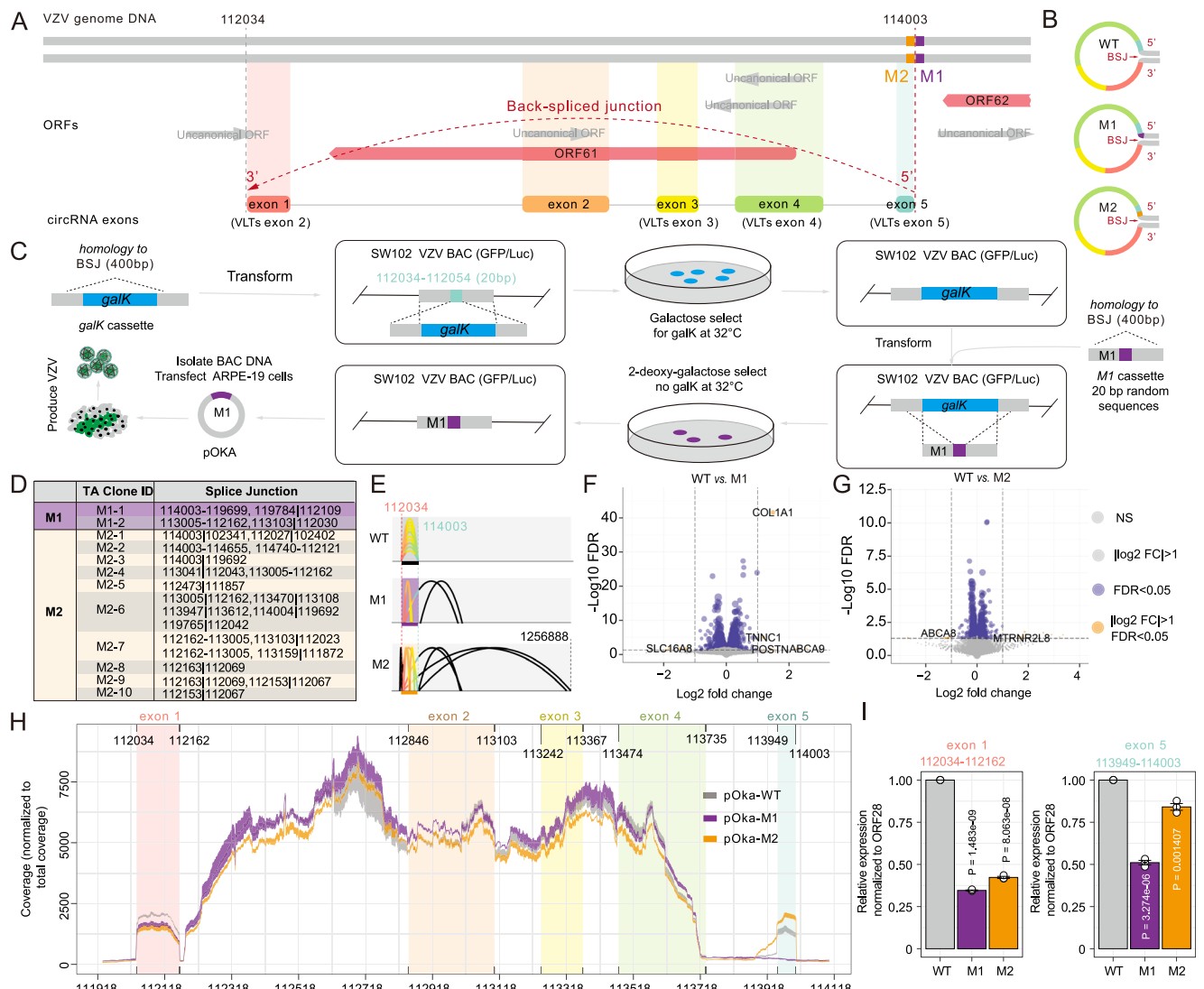

**Fig. 6 | Genomic level mutation of VZV circVLTs_lytic. A** Schematic diagram of sequence organization of the splicing event of VZV circVLTs_lytic at ORF61. The generation of the circVLTs_lytic BSJ exon 5 mutant position were marked as M1, purple and M2, orange. **B** The mutant sequences of M1, red and M2, black corresponding to the position of mature circVLTs_lytic. **C** Workflow of galK positive/counter selection VZV BAC system to generate VZV circVLTs_lytic BSJ exon 5 mutagenesis, named pOka-M1 and pOka-M2. **D** Statistics of Sanger sequencing results (Supplementary Data 4) generated mis-splicing BSJs of circVLTs_lytic 5' due to VZV BAC system mediated mutagenesis. "-" indicated forward-splicing junction, "|" indicated back-splicing junction. **E** Schematic diagram of the splicing location of mis-splicing BSJs of Sanger sequencing results. **F**, **G** Volcano plot shows differentially expressed cellular genes of pOka-M1 vs. pOka-WT (**E**) and pOka-M2 vs. pOka-WT (**F**). Log2 (fold change) is plotted as the abscissa, and log10 (FDR corrected P-value) is plotted as the ordinate, log10 (mean of read counts) as point size. **H** Genome coverage maps in the region of circVLTs_lytic of pOka-WT, pOka-M1 and pOka-M2. **I** Expression changes in the region of circVLTs_lytic exon1 (102816–102944) and exon 5 (104731–104785) of pOka-WT, pOka-M1 and pOka-M2. $N = 3$ independent experiments were performed. Statistical comparisons were made with two-sided unpaired t-test. The P-value of pOka-M1or pOka-M2 vs. pOka-WT group was shown. Data are presented as mean ± S.E.M. Source data are provided as a Source Data file.

identifying viral circRNAs. Importantly, numerous VZV circVLTs_lytic were discovered to be located in ORF61. VZV circVLTs_lytic contain multi-exon and multi-isoform VZV circRNAs within the same BSJ breakpoint. By mutating VZV circVLTs_lytic' genomic DNA location, we find that the sequence flanking the 5' splice donor is a cis-acting element for the formation of VZV circVLTs_lytic. Furthermore, viral growth curves showed that VZV circVLTs_lytic are dispensable for VZV replication, but the mutation downstream of circVLTs_lytic exon 5 led to acyclovir sensitivity in VZV infection models, including in vitro cultures of human skin tissue and human DRG, and in vivo xenografts of human skin tissue in SCID mice.

Several algorithms, including find_circ[31], CIRI2[32], CIRCexplorer2[42], circRNA_finder[43], Mapsplice[44], DCC[45], KNIFE[46], and UROBORUS[47] have been developed for the identification and quantification of circRNAs

based on BSJ reads from short-read RNA sequencing. Despite the computational methods available for short-read RNA-seq data to reconstruct full-length circRNAs such as CIRI-full[34], the inherent limitation of relatively short length reads does not experimentally determine the full-length sequences and isoforms splicing. Long-read sequencing technologies generate reads that are thousands to tens of thousands of bases in length, providing insights into transcriptome complexity for resolving full-length transcript isoforms. The ONT long-read sequencing, combined with RNase R treatment, rolling circular reverse transcription and corresponding software CIRI-long, experimentally provided method for reconstruction of full-length circRNAs. However, these algorithms often produce divergent results due to their distinct alignment methodologies and heuristics (Fig. 2J). To compare the performances of these circRNA algorithms, studies have

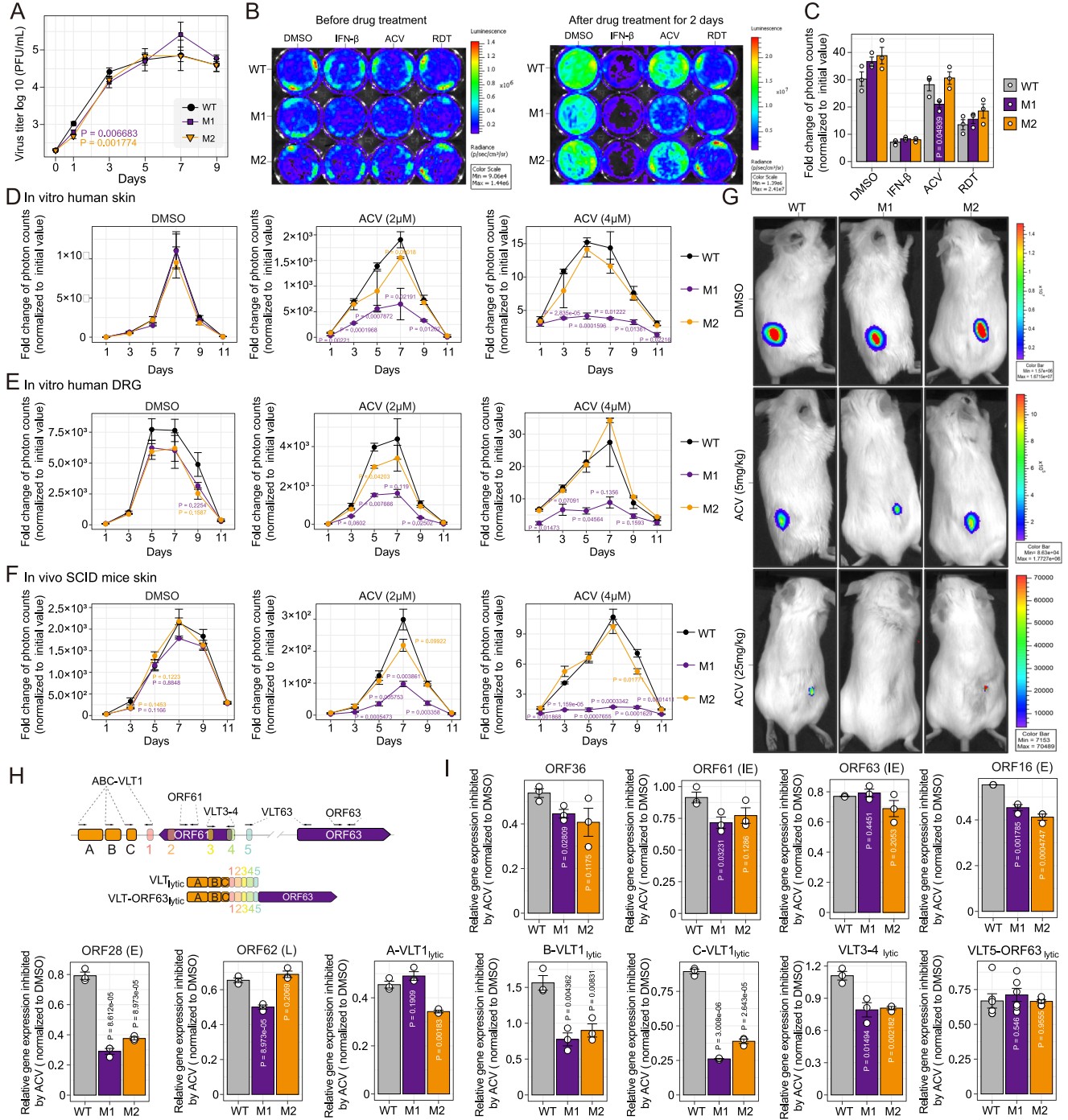

**Fig. 7 | Mutation of circVLTs_lytic exon 5 does not impair VZV replication but leads to acyclovir sensitivity.** **A** Growth curve of pOka-WT, pOka-M1, and pOka-M2 in ARPE-19 cells. VZV was titered by GFP fluorescent plaques. **B**, **C** Growth of pOka-WT, pOka-M1 and pOka-M2 treated with Interferons β (IFN-β, 100 u/ml), acyclovir (ACV, 2 μM/ml), rhodomyrtone (RDT, 1 μM/ml). Antiviral stress was added after ARPE-19 cells were inoculated with pOka-WT, pOka-M1 and pOka-M2 for 24 h. The virus replication levels were measured by adding D-luciferin and recording the bioluminescence signal with an in vivo fluorescence imaging system (IVIS) system. The average photon count values for each group were obtained from three independent experiments. The relative photon counts were measured by removing the initial value before IFN-β, acyclovir or RDT treatment (day 0) and then normalized to DMSO. **D–F** Statistical analysis of the total flux of VZV infection models, in vitro culture human skin (**D**), human DRG (**E**) with or without acyclovir (ACV, 2 μM and 4 μM) treatment, and in vivo xenografts of human skin tissue in SCID mice with or

without acyclovir (ACV, 5 mg/kg and 25 mg/kg) treatment (**F**). Statistical comparisons were made with one-way repeated analysis followed by LSD post hoc test. **G** Growth of pOka-WT, pOka-M1 and pOka-M2 in vivo xenografts of human skin tissue in SCID mice. **H** Schematic diagram of the primer location. Primers spanning VLTs splice junctions between exon 1 and upstream exons A-C for detecting VLTs were adapted from Ouwendijk, W. J. D. et al.[10]. Primers spanning VLTs splice junctions between exon 5 and ORF63 were adapted from Braspenning, S. E. et al.[11]. Primers for ORF61 and ORF63 adapted from our previous study Zhang, Z. et al.[9]. **I** Expression changes of ORF36, ORF61, ORF63, ORF16, ORF28, ORF62 and spanning VLTs splice junctions between exon 1 and upstream exons (**A–C**) and VLTs1-ORF63 of pOka-WT, pOka-M1 and pOka-M2 using qPCR. $N = 3$ independent experiments were performed. Statistical comparisons were made with two-sided unpaired t tests. The P value of pOka-M1 or pOka-M2 vs. pOka-WT group was shown. Data is presented as mean ± S.E.M. Source data are provided as a Source Data file.

utilized simulated datasets and RNA-Seq data with or without RNase R treatment to identify and quantify mammalian cellular circRNAs[48,49]. Among the algorithms tested, CIRI2 and KNIFE demonstrated better overall detection performance in RNA-Seq data with or without RNase R treatment[49]. It is important to note that RNase R treatment followed by RT-PCR and Sanger-sequencing is the gold standard for circRNA identification. In the present study, we conducted a comprehensive experimental screening on a genome-wide scale and identified a total of 200 VZV circRNAs from 235 clones (Supplementary Data 3). This experimentally identified VZV circRNA dataset serves as a robust foundation for the development of new circRNA algorithms.

To elucidate the function of a circRNA, overexpression or knock-down of the circRNA is often used[36]. For overexpression, a plasmid containing a circRNA along with its flanking intronic sequences is transfected into cells[50,51]. In a previous study, we attempted to over-express a SARS-CoV-2 circRNA, namely 29122|28295, in 293 T cells[28] using two established circRNA-expression vectors, pcD-ciR and pcDNA3.1 circRNA mini. However, our results showed that only a small fraction of expressed RNA was in circular form, making it challenging to study the function of circRNAs using this approach. Alternatively, RNAi or shRNA can be used to disrupt circRNA expression by targeting the BSJ site-mediated degradation[52,53]. However, it is challenging to design an RNAi with high efficiency to specifically target the shortened BSJ sequences, while avoiding off-target effects that could affect their residing genes. To overcome these challenges, our study employed two strategies: (1) we used the VZV BAC system to introduce mutations in the genomic DNA of VZV circVLTs$_{lytic}$. This allowed us to manipulate the circVLTs$_{lytic}$ at the DNA level; (2) To ensure that the observed effects were specific to circVLTs$_{lytic}$ and not influenced by other factors, we replaced 20 bp sequences upstream or downstream of the 5′ splice donor of circVLTs$_{lytic}$ with random sequences. Importantly, these sequences were carefully designed to avoid overlapping with any open reading frames (ORFs) in the viral genome (Fig. 6A).

In terms of circRNA functions, our study made significant progress and uncovered two interesting findings. Firstly, it has been reported that circRNAs regulate the expression of their parental genes[36]. In this study, we observed that certain VZV circRNAs are derived from essential genes, such as circRNA 3262|4061 and 3699|4061, which originates from ORF4. ORF4 is known to be essential for both VZV replication[54] and latency[55] (Fig. 3A). This suggests that these circRNAs may be involved in the regulation of essential viral gene expression. Secondly, we discovered a striking similarity between circVLTs$_{lytic}$ and VZV VLTs. These findings provide valuable insights into the functions of VZV circRNAs and their potential roles in viral gene regulation and infection. Further investigations are warranted to fully understand the precise mechanisms and functional implications of these circRNAs in VZV biology.

In terms of VZV pathogenesis, we identified VZV circRNAs from the tissues of HZ patients, which suggests a potential role of viral circRNA in VZV pathogenesis. However, further experiments are required to investigate the specific relationship between VZV circRNAs and the reactivation of VZV. While HZ is typically a self-limiting disease[56], antiviral therapies such as acyclovir, valacyclovir, and fam-ciclovir are used to reduce the risk of complications in immunocompromised patients[57]. Acyclovir, when present in infected cells, undergoes conversion by VZV thymidine kinase (TK), which is encoded by ORF36. The conversion produces acyclovir-monophosphate and further to acyclovir-triphosphate by cellular kinase enzymes, leading to the termination of viral DNA chains[58]. In our study, we demonstrated that mutation downstream of circVLTs$_{lytic}$′ 5′ splice donor site resulted in an increased sensitivity to acyclovir in VZV infection models, including ARPE-19 cells, in vitro culture human skin, human DRG, and xenografts of human skin tissue in SCID mice. These findings shed light on the potential role of circVLTs$_{lytic}$ in viral resistance and suggest that they may play a role in modulating the response to antiviral therapies.

However, it is important to note that ACV treatment rescued the expression of ORF36 from pOKA-M1, but there needs to be further exploration of whether circVLTs$_{lytic}$ exert their resistant effects on ACV through a mechanism that is distinct from function of TK.

Our findings revealed that the mutation in circVLTs$_{lytic,}$ which replaced 20 bases downstream (pOka-M1), resulted in an overall decrease in the expression of circVLTs$_{lytic}$. However, substitution of 20 bases upstream of circVLTs$_{lytic}$ (pOka-M2) only led to misalignment of circVLTs$_{lytic}$ with VLTs sequences (Fig. 6D−E). These results suggested that the sequence downstream of VLTs exon 5 serves as a cis-acting element essential for RNA splicing of circVLTs$_{lytic}$ and that circRNAs with VLTs sequences play a pivotal role in the different phenotypes between pOka-M1 and pOka-M2. During both latent and lytic infection, VZV expresses VLTs mRNA that encodes a protein, pVLTs[10]. VLTs are known to suppress the expression of ORF61 by acting antisense to its′ sequences independently of pVLT. The ORF61 is an immediate early gene, encoding the pORF61 protein that is a transcription factor regulating other VZV early (E) or late (L) genes and playing an important role in VZV replication[59,60]. Interestingly, we showed that the mutation of pOka-M1 exhibited a lower expression of ORF61, ORF28 and VLTs$_{lytic}$ than pOka-M2, suggesting that circRNAs with VLT sequences may play a pivotal role in viral resistance to acyclovir. Further investigations are needed to fully understand the specific mechanisms by which circVLTs$_{lytic}$ contribute to viral resistance and their relationship with antiviral therapies.

In conclusion, our current studies not only computationally identified VZV-encoded circRNAs but also experimentally validated the presence of VZV circRNAs in infected cells and tissues from HZ patients. The unprecedented identification of VZV circRNAs from the tissues of HZ patients suggests a potential significant role of viral circRNAs in VZV pathogenesis. Nevertheless, further experiments are required to investigate the relationship between VZV circRNAs and the reactivation of VZV. Additionally, our study delved into the role of interferons, such as IFN-α and IFN-β, as part of the cellular innate immune response to VZV infection. While HZ is generally self-limiting[56], antiviral therapies such as acyclovir, valacyclovir, and fam-ciclovir are used to mitigate complications in immunocompromised patients[57]. Notably, acyclovir undergoes conversion by VZV thymidine kinase (TK), encoded by ORF36, to acyclovir-monophosphate and further to acyclovir-triphosphate by cellular kinase enzymes, terminating viral DNA chain[58]. In this study, we demonstrated that the mutation downstream of circVLTs$_{lytic}$′ 5′ splice donor resulted acyclovir sensitivity in VZV infection. These findings shed light on the potential role of circVLTs$_{lytic}$ in conferring viral resistance.

## Methods

### Cell culture, virus and infection

ARPE-19 and SH-SY5Y human cell lines were gifts from Tong Cheng[61] (Development in Infectious Diseases, Xiamen University, Xiamen, China). ARPE-19 and SH-SY5Y cells were grown in DMEM - Dulbecco's Modified Eagle Medium (Gibco, USA, Catalog number: 10564029) with 10% fetal bovine serum (ExCell Bio, China, Catalog number: NCS500) and penicillin-streptomycin (100 U/ml and 100 g/ml, Gibco, USA, Catalog number: 15070063). Viruses, VZV pOka strain, containing GFP and luciferase coding genes was derived from the parental wild-type Oka strain. The genomic sequences of VZV pOka were assembled by whole genome sequencing which, the raw data were available in NCBI SRA (PRJNA1056528) and the complete genomic sequence was submitted to GenBank (Accession number, PP054841, https://www.ncbi.nlm.nih.gov/nuccore/PP054841).

### Ex vivo growth curve analysis of pOka, pOka-M1, and pOka-M2 in skin and dorsal root ganglion

Ex vivo growth curves of pOka, pOka-M1, and pOka-M2 were performed in ARPE-19 cells[8]. Briefly, human fetal tissues (skin and DRG,

18–20 weeks old) were obtained from Advanced Bioscience Resources (ABR). The tissues were washed with 1X PBS, processed, cultured in DMEM–F-12 (1:1) and 2% FBS and grown at 37 °C/5% $CO_2$[8,62]. Tissues were infected with pOka, pOka-M1, and pOka-M2 (100PFU/well). For drug treatment, 2 μM and 4 μM of acyclovir were added to the media. For luciferase activity measurement, 150 μg/ml of D-luciferin was added to the samples followed by incubation for 10 min at RT. Total photon counts were measured on using In Vivo Imaging System 50 (IVIS-50) and Xenogen (Live Image software).

### In vivo growth curve analysis of pOka, pOka-M1, and pOka-M2 in skin xenografts

Human fetal skin tissues (18–20 weeks old gestation) were obtained from Advance Biosciences Resources (Alameda, CA). The use of the human tissues in this study was approved by the Research Ethics Committee of Rutgers University. Animal experiments were carried out in accordance with all federal regulations in an AAALAC-accredited facility per the standards of the Guide for the Care and Use of Laboratory Animals and approved by the Institutional Animal Care and Use Committee (IACUC, No. 13-383) at Rutgers University, NJ, USA. The SCID mice were housed in SPF conditions with ad libitum access to food and water. The housing room was kept under controlled conditions: temperature ($23 \pm 2$ °C), 12 h dark/light cycle and ~50% relative humidity. The tissues were washed, processed, and subcutaneously engrafted in 4–6 weeks-old CB-17 SCID/beige mice[63]. In one set of experiments, three SCID mice in each group were infected with pOka, pOka-M1, and pOka-M2 ($4 \times 10^3$ PFU/mouse). The pOka, pOka-M1, and pOka-M2 growth curve analysis in engrafted tissues was determined by measuring the luciferase activity using IVIS-200 (Xenogen) on alternate days[61,63]. To check the virus's sensitivity to acyclovir, we used three SCID mice in each group and human skin was implanted subcutaneously. The implanted tissues were infected with pOka, pOka-M1, and pOka-M2 ($4 \times 10^3$ PFU/mouse) and treated with 5 mg/kg acyclovir. In another set of experiments, the infected mice were treated with 25 mg/kg of acyclovir to determine the virus's sensitivity to acyclovir. The acyclovir drug (5 mg/kg and 25 mg/kg) was injected intraperitoneally in the respective mice 8 h after viral infection, and luciferase activity was measured using IVIS-200 on alternate days.

### Total RNA extraction

SH-SY5Y cells were seeded at a density of $10^6$ in 6-well plate (Corning) 12 h prior to inoculation with log$10^6$ PFU of pOka. Cells were harvested at 24 and 48 h post-infection with TRIzol RNA Isolation Reagents (Invitrogen, USA, Catalog number: 15596026). Chloroform extraction and isopropanol precipitation methods were used to isolate RNA from samples. The extracted RNA was stored at −80 °C in a deep freezer before use.

### CircRNA sequencing for de novo circRNA identification and reconstruction

For short-reads circRNA sequencing, Ribosome-depleted and RNase R-treated RNA samples were used for library preparation and subsequently sequenced using the BGISEQ platform in PE150. The Gene Expression Omnibus database (GEO) accession number for the short-reads circRNA sequencing data reported in this paper is GSE223870. The analysis was performed on two Intel W-3175X CPUs with 128 GB memory running Ubuntu system (version 18.04). Raw reads were aligned with BWA Aligner[30] (BWA-MEM version 0.7.17-R1188) to human, VZV reference genomes: hg19, pOka strain (PP054841). Alignment statistics were performed with Qualimap2 (version 2.2.1)[64]. CIRI2 (version v2.0.6)[32] vircircRNA[25], and find_circ[31] were used for circRNA calling. Reconstruction of partial and full length circRNAs was performed with CIRI-full (version 2.0)[34]. Long-read circRNA sequencing was performed by Geneseed (Geneseed, Guangzhou, China). In brief, ribosomal RNA was depleted using RiboErase kit (KAPA RNA HyperPrep Kit with RiboErase for Illumina) and added A-tailing using Clean NGS (GC-Biotech B.V., Alphen aan den Rijn, the Netherlands) and subjected to RNase R treatment (Geneseed, Guangzhou, China). Rolling circular reverse transcription was performed using SMARTer PCR cDNA Synthesis Kit (Takara). The cDNA fragments with sizes within 200–2000 bp were selected for subsequent sequencing, using MinION (MN26543) platform (Nanopore Oxford). GEO accession number for long-reads circRNA sequencing data is GSE252124. CIRI-long (https://github.com/bioinfo-biols/CIRI-long) was used for full length circRNAs detection[16].

### RNA sequencing for gene expression quantification and RNA splicing analysis

RNA samples were used for library preparation and subsequently sequenced using the BGI DNBseq platform in PE150. The GEO accession number for RNA sequencing data reported in this paper are: pOka-WT, pOka-M1, and pOka-M2 infected-ARPE-19 cells (GSE223957). After removing the low-quality, adapter-polluted, and high content of the unknown base (N) read, the data for the clean reads were aligned to the human genome hg38 using STAR[65]. The counts of mapped reads for each gene were calculated using FeatureCounts from the sorted BAM files[66]. The differentially expressed genes (DEGs) were defined as genes with at least twofold changes between groups and an adjusted $p$ value less than 0.05 using DESeq2[33]. For VZV RNA splicing detection, RNA-seq raw reads were trimmed using trim_galore. Trimmed paired reads were pooled and aligned with the reference genomes using Python2 script ViReMa[67].

### Reverse transcription, inverse PCR and quantitative real-time PCR

About 4 μg total RNA or an equal amount of RNA treated by RNase R (Lecigen) was reverse transcribed with Prime Script TM RT Master Mix (Takara) using random hexamers. The divergent and convergent primers used in this study are summarized in Supplementary Data 3. Inverse PCR was performed with EasyTaq PCR SuperMix (TransGen Biotech) with 1 μl 1:20 diluted cDNA templated. Quantitative real-time PCR was performed with Tip Green qPCR Super Mix according to the manufacturer's instruction (TransGen Biotech).

### Cloning and identification of circRNAs

Inverse RT-PCR products were separated on 2% agarose gels. DNAs with the candidate BSJ sequences were gel purified (OMEGA Gel Extraction Kit) and TA-cloned (pMD18-T Vector Cloning Kit, Takara). At least 8 colonies were checked for the insertion of candidate PCR products by PCR with M13 universal primers. Following PCR purification (DNA clean & concentrator kit, Zymo), DNAs with candidate BSJ sequences were Sanger-sequenced (Tsingke Biotechnology Co., Ltd.) with M13 universal primers. Sequencing results were blasted against the VZV reference genome (pOka strain). 5′ and 3′ breakpoints of BSJs and FSJs were manually curated so that ambiguous nucleotides can be counted as the donor sequence if they exist around the junction. CircRNAs with BSJ breakpoints that differ within 20nt are considered as variants of one circRNA.

### Amplified fluorescence in situ Hybridization (AmpFISH)

AmpFISH probes to detect linear or circRNAs of VZV or HCR hairpins were designed, labeled, and purified[38]. The probe sequences are shown in Supplementary Data 3. Briefly, ARPE-19 cells were seeded on the glass coverslips (0.1 mm thickness), coated with 0.1% gelatin, and cultured in DMEM media with 10% FBS and 1% P/S. Cells were infected with Towne at an MOI of 0.1 and fixed at 72 h after infection with 4% paraformaldehyde/1XPBS for 10 min at RT. The cells were then washed with 1X PBS followed by incubation with 70% ethanol for 10 min at RT. For RNase R treatment, the cells were fixed with 4% PFA and permeabilized with 70% ethanol and 0.5% triton X100 for 10 min at RT. After

permeabilization, cells were then equilibrated with 1X reaction buffer (RNase R buffer) for 30 min, followed by incubation with 20 units of RNase R (Lucigen, RNR07250) for 2–3 h at 37 °C, and then, the amp-FISH procedure was applied. About 16 optical sections separated by 0.2 μm with 100–2000 milliseconds of exposure time were acquired for each image using 63x oil immersion objective in an Axiovert 200 M inverted fluorescence microscope (Zeiss, Oberkochen, Germany). The linear and circular RNA molecules in the infected cells were quantified by counting each RNA spot through an image processing program built in MATLAB (MathWorks, Natick, MA), as described earlier[38].

### Human clinical specimens and VZV clinical strain isolation

Human clinical samples in this study were approved by the ethics committee of Huazhong University of Science and Technology Union Shenzhen Hospital (Shenzhen Nanshan People's Hospital), Shenzhen, China. (No. 2016041201), allowing the publication of more than three identifiers. Written informed consent was obtained from all patients[39]. Blister fluid was obtained using a swab and stored at −80 °C. The rash tissue of three HZ patients was collected and and washed four times with DMEM medium, containing penicillin-streptomycin (centrifugation at 300 g for 5 min). After trypsin digestion for 25 min, the supernatant was co-incubated with ARPE-19 cells. Five days later, the VZV cytopathic effects were evaluated. The harvested cells with the clinical VZV strain was stored at −80 °C for future experiments. Genomic DNA sequences of the two VZV clinical strains were assembled using whole genome sequencing, and the genome sequence information was uploaded to Genebank under the Accession numbers PP261331 and PP261332.

### Generation of VZV circVLTs$_{lytic}$ mutagenesis using Bacterial Artificial Chromosome (BAC)

The upstream (113983–114003, from 5'-CCTGTAATACCCGTAAATAA-3' to 5'-TTCTAGTAGGATCGTTGCTG-3', Mutation2, pOka-M2) or downstream (114004–114024, from 5'-AGGTAAGTCCACAAACAAAA-3' to 5'-CCCACGATATCCCCTTAACT-3' Mutation1, pOka-M1) sequences of circVLTs$_{lytic}$ exon 5 were replaced with 20 bp random sequences from pOka genome via homologous recombination in the modified DH10B strain SW102 and a galK positive/counterselection cassette[29]. A whole-genome sequencing was performed to exclude off-target mutations. The corresponding recombinant virus was obtained by transfecting the ARPE-19 cells with the generated construct using the FuGene6 transfection kit (Roche, Indianapolis, IN).

### Plasmid construction and overexpression of circVLT

The DNA fragments of circVLT 419 nt and circVLT 542 nt isoforms were synthesized and cloned into a lentiviral circular RNA overexpression vector, pLCDH-ciR (Geneseed, Guangzhou, China)[68]. pLCDH-ciR, pLCDH-ciR-circVLT419, or pLCDH-ciR-circVLT542 and packaging plasmid psAX2 and pMD2G were transfected into 293 T cells using Lipofectamine 2000 (Invitrogen, Life Technologies, Carlsbad, CA, USA) for 48 h to produce lentiviruses. SH-SY5Y cells were incubated with lentivirues for 48 h, followed by puromycin (1 μg/ml) selection for 6 days to construct the cell lines expressing circVLT 419 nt and circVLT 542 nt isoforms.

### Reporting summary

Further information on research design is available in the Nature Portfolio Reporting Summary linked to this article.

## Data availability

All sequencing data used in this study have been deposited in the NCBI SRA and Gene Expression Omnibus database under accession code GSE223870, GSE252124, GSE223957 and PRJNA1056528. Source data are provided with this paper.

## Code availability

The code used in this study and extended data is available from the GitHub repository (https://github.com/ShaominYang/VZV_circRNA).

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

## Acknowledgements

This study was supported by grants from the Shenzhen science and technology innovations Committee (202108233000294) (L.X.), Science and Technology Project of Shenzhen Nanshan District Health System (No. NSZD2023034 and NS2024031) (S.Y.), Fujian Provincial Department of Science and Technology (No. 2020J05166) (J.L.) and Minnan Normal University Headmaster Fund (No. KJ18022) (J.L.), the National Institute on Minority Health and Health Disparities of the National Institutes of Health under Award Number G12MD007597 (Q.T.).

## Author contributions

S.Y., D.C., D.J., Q.T., and H.Z. designed the experiments, S.Y., D.C., D.J., M.W., and J.L. performed the experiments, S.Y., D.C., D.J., M.W., J.L., S.W., M.Z. and X.L. analyzed the data, S.Y., Q.T., D.X., L.X., M.Z. and H.Z. wrote the paper, Q.T., D.X., L.X., and H.Z. supervised the study.

## Competing interests

The authors declare no competing interests.
