## [Peer Review File · Nature Communications]

Identification and Characterization of Varicella Zoster Virus Circular RNA in Lytic InfectionREVIEWER COMMENTS

Reviewer #1 (Remarks to the Author):

Major points:

1. **Experimental Confirmation and In Silico Prediction:** The reviewer requests clarification on where in the manuscript the "15 high-confidence circRNAs" were experimentally confirmed (Page 7 Line 21). Additionally, there are concerns about the use of CIRI-full for de novo reconstruction of full-length viral circRNAs. The reviewer suggests that without long-read sequencing (LRS), the authors can only use CIRI-full as an in silico prediction and should accordingly present it as such.
2. **Mutation Impact on Protein and Transcript Expression:** The reviewer is interested to understand the impact of the 5' splice donor mutation of circVLT on the colinear gene product's protein. They request that the authors provide data to support the claim of absence of changes in transcript or protein expression from the coding sequence (CDS) in the circVLT region. The reviewer also questions whether the pOka-M1 and pOka-M2 mutations alter the protein-coding sequence of ORF61 and how the authors distinguish the effects of mutations impacting the circVLTs from a mutated ORF61 protein causing potential phenotypes.
3. **Bacterial Artificial Chromosome (BAC) Recombination:** The authors are asked to ensure there is no mutation of the viral thymidine kinase (or change in its expression) in the M1 virus before making the claim that circVLT impacts acyclovir resistance.
4. **Clarification on Figures:** The figures included in the manuscript require additional information to be fully interpretable. The reviewer requests specific details on Figures 2A, 2B-D, 3B, 5, S3B, and 8B. Also, it is noted that log frequency is not a common method for describing RNA levels of expression. Figure 2A. Is this plot only including reads where BSJ were detected? Or is this all reads from RNaseR sequencing that mapped to the genome? What is being plotted in Fig 2B-D? Is this the chimeric read junction? What are the axes and what is the vertical bar chart? What is the logFreq (log of normalized reads perhaps? If so, how were reads normalized)? In Fig 3B without labels it's hard to say, but it appears they are claiming stitching of multiple single exon VZV genes occurs to create circVLT variants? What RNA-Seq data is analyzed in Fig. 5?
5. **Phenotypic Evaluation:** The authors should provide an explanation of the phenotype in cells treated with ACV with only the M1 mutant (not M2), despite the claim that both mutants decrease circVLT expression. Also, the reviewer questions the normalization method and ask for the internal control showing the same number of cells in these control wells in Figure 8B.
6. **Host circRNAs:** The section on host circRNAs lacks functional insights, annotations, and interpretation. The reviewer suggests comparing host circRNA expression changes detected by the pipeline versus experimental result such as circHIPK3 (Figure 4C) to validate the bioinformatics.
7. **Validation of circVLT Isoforms:** More information and clarity is requested regarding the validation of full sequences of circVLT isoforms. Primer locations are unclear (Figure 4C-E), and full-length circRNA sequences are not included in the manuscript. The authors should consider conducting an orthogonal functional analysis to substantiate the loss-of-function claim, especially given the diversity of circVLT isoforms.
8. **Quantification and Replication of Results:** The reviewer asks for quantification of semi-quantitative RT-PCR results (Figures 4, S1, S3) and suggest using more quantitative methods such as real-time PCR and digital PCR for claims of up- and down-regulation. They also express the need for biological replicates in the results.
9. **Figure 8B:** There is more blue (low signal) in M1 and M2 with DMSO, compared to WT. However, graphs in the same figure panel show the same amounts of photon counts for all three conditions in DMSO. It appears that all three conditions are normalized to 1.0. Fig. S4A shows the same normalization to 1.0 for these conditions. There should be only one condition that is normalized to 1.0.

Minor comments:

1. The authors should clarify their VZV infection model in the text. We assume they chose a neuroblastoma line to mimic the latent reservoir of the virus, however cytopathic effects, and protein expression (Fig. S1) clearly indicate lytic infection. This is particularly misleading as the

- authors (Page 6 Lines 4-7), connect detection of the circVLT to regulation of VZV latency.
2. The statement on Page 8 Line 3-4 "The clinical identification of viral circRNAs from HZ patients suggests a potential correlation between viral circRNA and VZV reactivation-caused diseases." should be removed as it overreaches.
 3. Additional information should be provided for the analysis in Fig. 2 including:
 - The number of instances a viral BSJ variant was detected per sample
 - Overlap (reproducibility) of viral circRNA calls between biological replicates.
 4. To validate their RNaseR RNA-Seq claims of differentially expressed human circRNAs (Fig. 1) the authors would need to validate their findings in samples not treated with RNaseR by divergent qPCR or RNA-seq. Realistically they should also be normalized to spike-in controls to account for host shut off.
 5. Page 5 Line 6-7, please remove the line "potentially implicating them in the equilibrium of host immunity reaction and viral pathogenesis". You may discuss this possibility in the discussion, but merely identifying differentially expressed circRNAs during infection is not enough to make such a claim.
 6. Typo in Fig. 2b "find_cric" should be changed to "find_circ"
 7. Fig. 4B: "5' BSJ breakpoint hotspot" is not described in figure legend or text.
 8. Fig. 5 seems out of place and should be before the validation assays shown in Fig. 4.
 9. Fig. 7D: This table and legend are difficult to interpret. Please clarify for a broad audience.

Reviewer #2 (Remarks to the Author):

The manuscript by Yang et al represents the first ever analysis of VZV encoded circRNAs and by most measures is an impressive study. There are however a number of grey areas that I have elaborated on below in which more specific and detailed information is needed. I also have some concerns about the bioinformatics pipeline used and it's potential for introducing artefacts through relaxed filtering options and the structure of the VZV genome. I would also note that the authors are rather lax in citing recent VZV literature and as such refer in multiple situations to outdated concepts regarding VZV latency and the known coding content of the genome. While my list of concerns is rather long, I do think they should all be possible to address and I hope the authors will agree that they would significantly strengthen this interesting study.

Major concerns:

1. The introduction is rather lightweight and outdated. For instance, it has been shown that during VZV latency, only a single transcript (VLT) is expressed (PMID: 29563516, PMID: 22740396), rather than the 6+ transcripts indicated by the authors. Similarly, the authors fail to cite a recent transcriptome annotation exercise (Braspenning et al 2020 PMID: 33024035) that provided a far clearer picture of the VZV transcriptome.
2. Despite a general lack of spliced transcripts across the VZV genome (exceptions including ORF0, ORF50, and VLT), the authors noted over 185 distinct VZV circRNAs. This seems excessive and presumably results from very lax filtering parameters for back-splice junctions (counts ≥ 2). It would be helpful to have far more detail on the distributions of these counts (e.g. BSJ counts vs genome position) and some further analysis to indicate how the results would change with stricter count filtering (e.g. counts $\geq 10, 50, 100$). The authors should also indicate how many of the discovered circRNAs are novel and how many have been previously annotated in, for instance, circBank (<http://www.circbank.cn>).
3. Relating to this and upon inspecting the provided code, I note that the authors use BWA mem in their alignments but perform separate alignments against the human and viral genomes. As BWA mem is a local aligner, this often leads to misalignments that can increase noise and artefact in the resulting data. The authors should instead perform a competitive alignment i.e. append the viral genome to the human genome FASTA and then align the sequencing data (as indeed the authors performed for the RNA-Seq analysis).

4. The specific regions enriched for putative VZV circRNAs (line 11, page 5) are interesting but the accompanying figure lacks any detail on the genomic structure and coding content. While hard to estimate from the plot, the peak around 11k might coincide with the R1 repeat region while the peaks at the right-hand end of the genome look to co-localize with the duplicated IRS and TRS repeats. Can the authors show these peaks are not simply artefacts aligning occurring from alignment to the repeat regions? In other words, did they allow for reads to map to multiple locations and if so, did they filter to only retain a single alignment per read?

5. The genome-wide mutagenesis library would appear to be missing important spliced RNAs such as VLT and VLT63 which is a significant oversight. Moreover, it would be important to determine whether VZV circRNA biogenesis is significantly impacted by VLT/VLT63 deletion (or replacement with cDNA copies).

6. The authors should delete the line suggesting that VLT-associated circRNAs may play a regulatory role in latency as there is absolutely no data to show these are even made in latently-infected neurons, let alone serve a function (line 6, page 6)

7. I'm afraid I'm rather lost on how the systematic experimental genome-wide scanning (line 10, page 7) was performed, let alone how this gave rise to 185 VZV circRNAs. The authors need to elaborate on this in both the results and methods sections.

8. The authors should delete the line suggesting that VZV circRNAs have anything to do with VZV reactivation caused diseases, again due to a lack of any data whatsoever supporting this.

9. The authors are incorrect to state that pVLT suppresses expression of ORF61 as the original study demonstrated this was mediated at the VLT RNA (and not protein) level (line 4, page 12). Moreover, pVLT is not detectable during latent infection and the studies relating to ORF61 expression were carried out using expression vectors that would not encode VLT circRNAs, making this whole line of speculation incorrect.

Minor concerns

1. The authors align against the wildtype pOka genome but are using a mutant pOka strain containing GFP and luciferase. Ideally, all alignments should be against this specific genome to reduce mismapping issues. It is also not described in the manuscript where the luciferase insertions. Ideally, a genome sequence for this mutant would be available and deposited at NCBI.

2. I would argue against the use of the term circVLT as this could be interpreted as circRNAs that are expressed during viral latency (for which there exists no evidence to date).

3. The authors mentioned pooling biological replicates but I did not see any evidence provided that their biological replicates were sufficiently similar to do so.

4. Did the authors confirm the mutations (and absence of off-target mutations) in pOka-M1 and pOka-M2 by whole genome sequencing? This would be important to understand the differing results between M1 and M2.

Reviewer #3 (Remarks to the Author):

In this manuscript, "Identification and Functional Characterization of Varicella Zoster Virus Circular RNA in Lytic Infection," the authors performed sequencing on libraries made from RNase R treated RNA samples of VZV-infected materials. VZV-derived circular RNAs were identified through computational circRNA prediction algorithms, and VZV latency-associated transcripts (VLT)-like circRNAs were targeted for experimental validation and functional studies. For the latter a BAC

system was used to disrupt the expression of the circVLT by mutating sequences flanking the circVLT's splice donor site. It was concluded that disruption of the circVLT did not affect virus replication, but lead to increased acyclovir sensitivity in in vitro culture of human skin tissues and a SCID mouse xenograft model.

Generally, the computational identification of VZV circular RNAs is an advance in the field by defining a new group of RNA elaborated by VZV. However, the experimental studies of the VZV circular RNAs do not sufficiently support the authors conclusions.

With respect to experimental confirmation of the identity of VZV circRNAs. Several things are requested:

1. Figure 2A: This figure only shows genome coverage with RNase R digested RNA samples from VZV-infected materials. VZV coverage with RNA not treated with RNase R should also be included/overlaid for comparison to rule out any RNA sequence bias. The peaks identified should be scaled by ORF boundaries
2. Figure 3A: It would be helpful if peaks identified in Fig. 2A can be related to the putative circular RNAs identified. Do they peaks and circRNAs overlap significantly—do they correspond meaningfully. How are the reconstructions of the "Full" circular RNAs obtained? Is this by divergent primer PCR followed by sequencing. It is critical to state whether this is done by alignment of reads from initial sequencing or by subsequent experimental confirmation.
3. Figure 4C: The labeling of the bands (red arrowheads and number) completely obliterates whether weak bands are seen in mock lanes at corresponding positions. P3 looks not viral, since band appears in mock lane – similar comment on P8. Were these PCR products sequenced? Since the authors are concentrating in circVLTs and full map of isoforms that are sequence confirmed is needed.
4. Figure 6: Bands identified as circRNA from the 6 different patients need qPCR done with RNase R treated and untreated conditions, and sequenced to confirm they are predicted backspliced amplicons.

With respect to the assertion that disruption of circVLT confers increased acyclovir sensitivity. Fussing with any viral sequence requires detailed examination of "off-target" effects.

1. Does the mutation introduced into the BAC include any part of the canonical splice sequence?
2. The changes in circVLT isoforms as well as linear transcript need to be enumerated and documented.
3. Does introduction in trans of the linear VLT reconstitute wild-type phenotype?

REVIEWER COMMENTS

We appreciate the time and the effort the reviewers have invested in reviewing our manuscript and the feedback is invaluable to us. We have diligently conducted the experiments and analyses the reviewers suggested, and we thoroughly addressed all the concerns the reviewers raised. We believe these enhancements contribute to the robustness of our results and the overall quality of the manuscript. Once again, we express our gratitude for the thoughtful review.

Reviewer #1:

Major points:

1. Experimental Confirmation and In Silico Prediction: The reviewer requests clarification on where in the manuscript the "15 high-confidence circRNAs" were experimentally confirmed (Page 7 Line 21). Additionally, there are concerns about the use of CIRI-full for de novo reconstruction of full-length viral circRNAs. The reviewer suggests that without long-read sequencing (LRS), the authors can only use CIRI-full as an in silico prediction and should accordingly present it as such.

Response 1-1:

We are grateful for the reviewer's valuable questions, particularly suggesting the LRS. In this study, we initially identified a large number of circRNAs using various circRNA analysis algorithms. Subsequently, through extensive experimental validation, a total of 200 circRNAs were confirmed from 235 clones. To avoid potential misunderstandings, we have removed Figure 5, and the relevant information will be discussed separately in Figure 2 and Figure 4.

To enhance the detection and reconstruction of full-length circRNAs, we supplemented our approach with the long-read sequencing (LRS) of circRNA based on Oxford Nanopore Technologies (ONT), combine with RNase R treatment and rolling circular reverse transcription, following the method proposed by Zhang J (Zhang J, Nat Biotechnol, 2021, PMID: 33707777). Please refer to Figure 2 and Figure 3 for details.

2. Mutation Impact on Protein and Transcript Expression: The reviewer is interested to understand the impact of the 5' splice donor mutation of circVLTs on the colinear gene product's protein. They request that the authors provide data to support the claim of absence of changes in transcript or protein expression from the coding sequence (CDS) in the circVLT region. The reviewer also questions whether the pOka-M1 and pOka-M2 mutations alter the protein-coding sequence of ORF61 and how the authors distinguish the effects of mutations impacting the circVLTs from a mutated ORF61 protein causing potential phenotypes.

Response 1-2:

Thanks for reviewer's important questions that ensure our results' accuracies.

As shown in the figure below: in terms of experimental design, we minimized potential interference factors by avoiding changes in VZV ORFs, especially ORF61. We introduced a 20-base substitution only at the splicing site (114003) upstream (pOka-M1) or downstream (pOka-M2) of the fifth exon of circVLTs_{lytic}. Therefore, our mutations will not affect the gene expression of ORF61.

Since the sequence of circVLTs_{lytic} is derived from VLTs, the fifth exon of circVLTs_{lytic} is also the fifth exon of VLTs. Therefore, there will be a slight impact on the fifth exon of VLTs: 6 amino acids

in the C-terminus of circVLTs_{lytic} CDS in pOka-M1; no change occurs in the circVLTs_{lytic} CDS of pOka-M2.

In this revised version, we supplemented the whole-genome sequencing data (the data has been uploaded to NCBI SRA, <https://www.ncbi.nlm.nih.gov/sra/PRJNA1056528>) for pOka-WT, pOka-M1, and pOka-M2 and assembled their genome sequences, confirming that BAC recombination only affected the 20-base sequence upstream or downstream of the splicing site (114003) of the fifth exon of circVLTs_{lytic}.

In addition, we confirmed that the mutations in pOka-M1 and pOka-M2 do not affect the splicing of VLTs and VLT63 (Figure 7H-I).

ACGGACTTACCAGGGGGCAGTATTTACACCTTGGGTTCCAGATATACCAACCCTTACGACCAATAGCAACACTCAGGTATTTTTAAAAATGCAGTTTAATGATCATAATTTACATACAGTTGGTA
ATAAAGCAGACTGTGGATGTTAAAGCATTTCCTTCCCCTCCCAACAAA-[ORF61CDS CTAGGACTTCTTCATCTTGTGGAAATACCTTTACCOCGTTTACCOCGAGAGCTTTTTT
TGTAAGGTGTTTCAGTGAACCTGATGTTGATCCGGAGGTGGAGGGGATTGGACTCCCCTGTGGAGAGGCAACTTTGCCGGTTTTACTTCCCTTACATGCCGAATCAGACTCAGATGTCA
GGTCTATTGTTAAGCATCGTTTAAAGCTCTCTGCCGGTATGAAATAAACGGCGCTTAGCACCCCTTTCGCTTCCCGTTTTAATCCCGGTAACACAGAAAAAGCCTGACTTTTTGGGGTGTATT
TACCAATCCGGTATCCCTTTCATCGCCACGAGAGGTCTCCCGGTTGAGGTGGTTCTGGTCTTACAATTGGACCTGTAATTAGTTGGATGGCTGTATCTTCCAGGTCCAGGTTTGCATGGTT
AGGCGGGTTGGATCGGTACATCGATCCAACAAGAAATAACATGTTTGTACAAAGGGTCTGTGTAATCATGCAAAAAGACAACGCAGGGATGTTTTAATCCCGCTCATCACGCCGTAATAC
CTATATAGTTAATCAACATTTTTGTAGGCTCTACAATTTCCGGTTGATACAGTTCCGCAAGTTGATCATCAAGCCATCCGAGTAAAGGTTGCATGTAAACACGGGAATCTCCGCTTCCCTCT
GTTCCCTATCCGGTCCGAAAAGGCAGTCTGTCCATGGTTCGGTGGTCTTGATTAATCCCCACAGATACGGAGTACCGGTAGTCTGCCCCCGGTCGGGGTTGCTGTGCAGATTCC
AATCGAGCCATACACCACCCGGGTCGCCGATCGAACAGCAGGTTGGTCTTAAAAAATACCCTCCGTAATAATGATGCGGTAGAGCATGTTTTGGTTACACCAGGCTCGAGTCTCGGGTCG
GTGGTTGTATAGAATCCTGTTGAGAGTCACTTGGTACTCTGCTGTGGGCTCTAGCCGACGATTGAAGGGGCCAGGGTTTGGTATTGAATGGGCTCCCGACTCGATCTGATGTTGGC
TGTTGGATGGACTCCCGACTCCGCTCGGCTTGGTGGCAGAAGATCTATGACATCTCCCGTAGGATGTCGATGGAATCTTCAAATGACGGCTCAGAAAAACCATCGTCCGTCGGATGGGT
CACTTCATATCTTGTAACTTGTATACACTTACGATCTTATGCAGGATGGATTGCACTGGACACCCGGCAGAGAGGACACTGGACGCTGGTGGAGGTTCCATGCCCGAATAACAAACAAAGCAGAA
GTCGTGCAAAACACGGCATGGTTTTCCGAGATCGGAAACGGTGCATGCATATGGTGCAGGATTATCCGAAGCGTCGGAGGTGCCGCTACCOCGCTAATATGGTATCCAT]-GGTAAC
AACTGGCTGATTCTAATGTCGGGCATCCAACACGTAGCAGAAGTCCATCGCTTCTAAATGTGAGTTGGCGAGTACATTTTTATAATGGTACCAACGAAGACACACCCCTATATCCC
TCCACCACTTCTTTAAGTCCACCCACTAAAACGTGGGTATAAAATGTGATTGGGTAGGCGGACAGTCCCAACAAACAGGGAAGTTGATTGGTATAACCTTGGCCGGGTATACAGCTA
AGTGACATTTTAGATTCTGTCTTTATTAGATAAAGAGCGATACGAAGACATTTCTCCACCCC-¹¹³⁹⁸³CCTGTAATACCCGTAATAATA¹¹⁴⁰⁰³-¹¹⁴⁰⁰⁴GGT
AAAGTCCACAAACAAA¹¹⁴⁰²⁴-GCACTGTATATAGGAAGTCGGGTGATTGGGACAGTTACTCCATTAGAGGCGTACAA

pOka-M2: from CCTGTAATACCCGTAATAA to TTCTAGTAGGATCGTTGCTG
pOka-M1: from AGGTAAGTCCACAAACAAA to CCCACGATATCCCCTTAACT

> CDS in the pOka-WT circVLTs_{lytic}
MPRLLRDRIAGIPNVRVRYQGAVFTPWVPDIPTLTNSNTQILDDHGSAPRSGVAVQIQSSHTPPGSPIEQQDGLHWTPAERTLDAG
GGPCPNTNKAIEVVQTRHGFSEIGNAHAYGADKERYEDISPPPCNTRK*

> CDS in the pOka-M1 circVLTs_{lytic}
MPRLLRDRIAGIPNVRVRYQGAVFTPWVPDIPTLTNSNTQILDDHGSAPRSGVAVQIQSSHTPPGSPIEQQDGLHWTPAERTLDAG
GGPCPNTNKAIEVVQTRHGFSEIGNAHAYGADKERYEDISPP**LLVGSLLR***

> CDS in the pOka-M2 circVLTs_{lytic}
MPRLLRDRIAGIPNVRVRYQGAVFTPWVPDIPTLTNSNTQILDDHGSAPRSGVAVQIQSSHTPPGSPIEQQDGLHWTPAERTLDAG
GGPCPNTNKAIEVVQTRHGFSEIGNAHAYGADKERYEDISPPPCNTRK*

3. Bacterial Artificial Chromosome (BAC) Recombination: The authors are asked to ensure there is no mutation of the viral thymidine kinase (or change in its expression) in the M1 virus before making the claim that circVLT impacts acyclovir resistance.

Response 1-3:

Firstly, it is crucial to note that the viral thymidine kinase (TK) gene ORF36 is distantly located from the mutation site. Our comprehensive analysis involved confirming that the BAC mutation had no impact on the encoding frame of ORF36 genomic DNA. This verification was conducted through whole-genome sequencing and Sanger sequencing of the BAC DNAs from pOka-WT, pOka-M1, and pOka-M2.

Secondly, the examination of the full-length mRNA of ORF36 in pOka-WT, pOka-M1, and pOka-M2 aimed to ensure that the BAC mutation did not affect the encoding frames of ORF36. This confirmation is achieved through RT-PCR and Sanger sequencing.

Interestingly, qPCR analysis of the ORF36 expression in pOka-WT, pOka-M1, and pOka-M2 revealed lower expression levels of ORF36 in pOka-M1 and pOka-M2 compared to that in pOka-WT. Therefore, the mechanism rendering pOka-M1 sensitive to ACV need to be further explored in the future.

4. Clarification on Figures: The figures included in the manuscript require additional information to be fully interpretable. The reviewer requests specific details on Figures 2A, 2B-D, 3B, 5, S3B, and 8B. Also, it is noted that log frequency is not a common method for describing RNA levels of expression. Figure 2A. Is this plot only including reads where BSJ were detected? Or is this all reads from RNase R sequencing that mapped to the genome? What is being plotted in Fig 2B-D? Is this the chimeric read junction? What are the axes and what is the vertical bar chart? What is the logFreq (log of normalized reads perhaps? If so, how were reads normalized)? In Fig 3B without labels it's hard to say, but it appears they are claiming stitching of multiple single exon VZV genes occurs to create circVLTs variants? What RNA-Seq data is analyzed in Fig. 5?

Response 1-4:

We have supplemented detailed descriptions for original Figures 2A, 2B-D, 3B, 5, S3B, and 8B, along with other images.

The original Figure 2A depicted the genomic coverage of all viral RNA reads mapped to the viral genome. To analyze the resistance of VZV-transcribed RNA to RNase R, we have now included genomic coverage data for both RNase R-digested and undigested samples comparatively.

In this manuscript, to better capture the features of VZV-encoded circRNAs, we used three different algorithms to analyze short sequencing data (Figures 2G-I) and CIRI-long to analyze long sequencing data (Figures 2J). The coordinates (Start | end) of each circRNA in the genome are shown on the x and y axes of Figures 2G-J, with colors representing the number of reads on the supporting circRNA BSJs. To make different software more comparable quantitatively, we normalized the supporting BSJ reads of each circRNA by dividing it by the total BSJ reads of all circRNAs identified by that software, i.e., $\log \text{Frequency} = \log (\text{BSJ reads of each VZV circRNA} / \text{Total VZV BSJ reads})$.

As the reviewer speculated, Fig 3B (now Fig 3C) demonstrates the generation of circVLTs_{lytic} variants through multiple circexon splicing. Additional labeling information has been provided.

The data in the original Fig. 5 included circRNAs identified by various prediction algorithms and experimentally validated circRNAs. Because this section could lead to misunderstandings, we have now removed Figure 5 and explained the findings separately in Figures 2 and 4.

5. Phenotypic Evaluation: The authors should provide an explanation of the phenotype in cells treated with ACV with only the M1 mutant (not M2), despite the claim that both mutants decrease circVLT expression. Also, the reviewer questions the normalization method and ask for the internal control showing the same number of cells in these control wells in Figure 8B.

Response 1-5:

We would like to thank the reviewer for the valuable suggestion. We have addressed the concern regarding the phenotype in cells treated with ACV with only the M1 mutant (not M2) by providing a detailed explanation in the discussion section of the manuscript (Page 14 lines 27).

Regarding the normalization and internal control for cell numbers, our approach is as follows. Cells were inoculated in a six-well plate with the same number, and upon reaching 90% confluence they were infected with the three viruses (pOka-WT, pOka-M1, and pOka-M2) for 24 hours. This ensures that the number of cells in all wells was consistent. The bioluminescence signal for control was recorded before antiviral treatment (Figure 7B). Despite efforts to infect cells with same viral titer, slight differences may exist. To minimize these variations, corrections were implemented. In our correction method, we used the fold change of virus growth normalized to the initial value. Additional details have been included the figure legend to provide a clear explanation of the normalization procedure.

6. Host circRNAs: The section on host circRNAs lacks functional insights, annotations, and interpretation. The reviewer suggests comparing host circRNA expression changes detected by the pipeline versus experimental result such as circHIPK3 (Figure 4C) to validate the bioinformatics.

Response 1-6:

In response to the reviewer's suggestion, we did the following analysis to strengthen our manuscript. Firstly, we aligned the identified cellular circRNAs with the CircBank database and found that the majority of highly expressed circRNAs are included in the database (Figure 1A-B and Data S1). Secondly, we conducted preliminary functional predictions for the genes from which circRNAs originate (Figure 1J-K). Finally, we selected up-regulated and down-regulated circRNAs for validation through qPCR and Sanger sequencing (Figure 1H-I).

7. Validation of circVLT Isoforms: More information and clarity is requested regarding the validation of full sequences of circVLTs isoforms. Primer locations are unclear (Figure 4C-E), and full-length circRNA sequences are not included in the manuscript. The authors should consider conducting an orthogonal functional analysis to substantiate the loss-of-function claim, especially given the diversity of circVLTs isoforms.

Response 1-7:

We have supplemented the schematic diagram to show the divergent primer positions and the Sanger sequencing results for validation of full-length circVLTs_{lytic} (Figure 4C-E). The full-length sequence of circVLTs_{lytic} is provided in Data S3.

As shown in Figure 3B-C, we have analyzed different isoforms of circVLTs_{lytic}, most of which possess the second and fifth exons of VLTs. Our mutation site is located upstream (pOka-M1) and downstream (pOka-M2) of fifth exons of VLTs, affecting all these isoforms. As indicated in Figure S4B, Figure 6D-E and Data S4, we have analyzed the impact of the mutation in the upstream and downstream regions of the fifth exon of circVLTs, suggesting a possible association between sensitivity to acyclovir and the presence of the VLTs sequence. For details, please refer to **Response 1-5**.

8. Quantification and Replication of Results: The reviewer asks for quantification of semi-quantitative RT-PCR results (Figures 4, S1, S3) and suggest using more quantitative methods such as real-time PCR and digital PCR for claims of up- and down-regulation. They also express the need for biological replicates in the results.

Response 1-8:

In response to the suggestion, we have added quantification of the bands of three biological replicates in both Figure 4B and Figure S3C, showing in Figure S3D-E and Figure S1B-C.

9. Figure 8B: There is more blue (low signal) in M1 and M2 with DMSO, compared to WT. However, graphs in the same figure panel show the same amounts of photon counts for all three conditions in DMSO. It appears that all three conditions are normalized to 1.0. Fig. S4A shows the same normalization to 1.0 for these conditions. There should be only one condition that is normalized to 1.0.

Response 1-9:

In this revised version, we have implemented a normalization strategy based on the fold change of virus growth. This involves normalizing the photon counts of each sample to its initial value before antiviral treatment. This approach serves to mitigate the potential impact of experimental errors in the infection MOI of the three viruses, as depicted in Figure 7B. By employing this normalization method, we aim to ensure the robustness and reliability of our results, taking into account any variations in the viral titers during the experimental procedures. For details, please refer to **Response 1-5**.

Minor comments:

1. The authors should clarify their VZV infection model in the text. We assume they chose a neuroblastoma line to mimic the latent reservoir of the virus, however cytopathic effects, and protein expression (Fig. S1) clearly indicate lytic infection. This is particularly misleading as the authors (Page 6 Lines 4-7), connect detection of the circVLTs to regulation of VZV latency.

Response 1-10:

We thank the reviewer for pointing out this important issue. Indeed, this study only investigated the lytic infection of VZV. Following the approach of Depledge DP, et al. (Nat Commun, 2018, PMID: 29563516), we have clarified in the manuscript that it is circVLTs_{lytic} to avoid this misunderstanding.

2. The statement on Page 8 Line 3-4 "The clinical identification of viral circRNAs from HZ patients suggests a potential correlation between viral circRNA and VZV reactivation-caused diseases." should be removed as it overreaches.

Response 1-11:

It has been removed.

3. Additional information should be provided for the analysis in Fig. 2 including:

- The number of instances a viral BSJ variant was detected per sample
- Overlap (reproducibility) of viral circRNA calls between biological replicates.

Response 1-12:

The reviewer's suggestion is very helpful.

We have added Figure S2B-H to show the identification of VZV circRNA number in each biological replicate sample. The "pool_all" represents the VZV circRNA identified by combining all the sequenced raw data treated with RNase R for bioinformatics analysis.

As seen in the heatmap (Figure S2F-H), circRNAs on the sides of the exterior are identified in most samples. Due to the generally low expression abundance of circRNAs, merging all samples (increasing sequencing depth) allows for the identification of more VZV circRNAs (highlighted in purple).

4. To validate their RNaseR RNA-Seq claims of differentially expressed human circRNAs (Fig. 1) the authors would need to validate their findings in samples not treated with RNaseR by divergent qPCR or RNA-seq. Realistically they should also be normalized to spike-in controls to account for host shut off.

Response 1-13:

We selected circRNAs from cells that are both included in the circbank database and regulated by VZV infection for validation through Sanger sequencing (Figure 1L). For samples with and without RNase R digestion, the expression trend of these circRNAs, assessed using linear RNA (GAPDH) as an internal reference through divergent qPCR results, was found to be consistent with the bioinformatics analysis (Figure 1M).

5. Page 5 Line 6-7, please remove the line "potentially implicating them in the equilibrium of host immunity reaction and viral pathogenesis". You may discuss this possibility in the discussion, but merely identifying differentially expressed circRNAs during infection is not enough to make such a claim.

Response 1-14:

As the reviewer suggested, we have removed it.

6. Typo in Fig. 2b "find_cric" should be changed to "find_circ"

Response 1-15

We have corrected this error in the revised MS.

7. Fig. 4B: "5' BSJ breakpoint hotspot" is not described in figure legend or text.

Response 1-16:

Corrected

8. Fig. 5 seems out of place and should be before the validation assays shown in Fig. 4.

Response 1-17:

Figure 5 has been integrated into Figure 2 and Figure 4, as the reviewer suggested.

9. Fig. 7D: This table and legend are difficult to interpret. Please clarify for a broad audience.

Response 1-18:

We have revised the legend to more clearly describe the table.

Reviewer #2:

The manuscript by Yang et al represents the first ever analysis of VZV encoded circRNAs and by most measures is an impressive study. There are however a number of grey areas that I have elaborated on below in which more specific and detailed information is needed. I also have some concerns about the bioinformatics pipeline used and it's potential for introducing artefacts through relaxed filtering options and the structure of the VZV genome. I would also note that the authors are rather lax in citing recent VZV literature and as such refer in multiple situations to outdated concepts regarding VZV latency and the known coding content of the genome. While my list of concerns is rather long, I do think they should all be possible to address and I hope the authors will agree that they would significantly strengthen this interesting study.

Response: We would appreciate the reviewer's comments and suggestions that are all helpful in improving our manuscript. We have performed experiments and analyses to address all the reviewer's concerns and questions.

Major concerns:

1. The introduction is rather lightweight and outdated. For instance, it has been shown that during VZV latency, only a single transcript (VLT) is expressed (PMID: 29563516, PMID: 22740396), rather than the 6+ transcripts indicated by the authors. Similarly, the authors fail to cite a recent transcriptome annotation exercise (Braspenning et al 2020 PMID: 33024035) that provided a far clearer picture of the VZV transcriptome.

Response 2-1:

We are grateful to the reviewer's critical comments and we have modified the text. We have cited the important references, and we also added more information regarding the background of VLTs with more updated information and references.

2. Despite a general lack of spliced transcripts across the VZV genome (exceptions including ORF0, ORF50, and VLTs), the authors noted over 185 distinct VZV circRNAs. This seems excessive and presumably results from very lax filtering parameters for back-splice junctions (counts ≥ 2). It would be helpful to have far more detail on the distributions of these counts (e.g. BSJ counts vs genome position) and some further analysis to indicate how the results would change with stricter count filtering (e.g. counts $\geq 10, 50, 100$). The authors should also indicate how many of the discovered circRNAs are novel and how many have been previously annotated in, for instance, circBank (<http://www.circbank.cn>).

Response 2-2:

We thank you the reviewer for valuable suggestions and careful reading of our MS.

VZV lytic infection has been reported to generate a large number of transcript splicing events (Prazsák I, BMC Genomics. 2018. PMID: 30514211), where they identified 114 novel viral transcripts. In our previous work, we discovered 351, 224, and 2764 viral circRNAs in cells infected with SARS-CoV-2, SARS-CoV, and MERS-CoV, respectively (Yang S, 2022, J Med Virol, PMID: 35318674). In another alpha-herpesvirus, HCMV, we identified 324 viral circRNAs through

TA cloning (Yang S, 2022, Microbiol Spectr, PMID: 35604147). In this experiment, we enriched circRNA using RNase R followed by deep short-reads sequencing and long-reads sequencing. Bioinformatic analyses identified 106 (CIRI2), 2806 (find_circ), 305 (vircircRNA), and 1358 (CIRI-long) viral circRNAs, which were further confirmed by TA cloning through PCR, resulting in the identification of 200 viral circRNAs. This suggests that the production of a large number of circRNAs by viruses may be a common phenomenon.

Currently, most circRNA algorithms identify one circRNA with one BSJ supported read. Following the criterion proposed by Zhao et al., we identify a circRNA with at least two BSJ supported reads. We have reconstructed the full-length of cellular and viral circRNAs using two sequencing platforms, and we have supplemented Figure 2 B-E to show the size of full-length circRNAs corresponding to their expression levels. It can be observed that the majority of VZV circRNAs have expression BSJ reads greater than 10.

The suggestions from the reviewer have been very instructive. The circBank database primarily catalogs circRNAs from animals and plants, with relatively fewer entries for viral circRNAs. Inspired by the reviewer's suggestion, we have annotated cellular circRNAs using circBank (Figures 1 and Data S1). We noted the existence of a database, ViruscircBase, dedicated to viral circRNAs, which includes 78 VZV circRNAs. However, the credibility of the VZV circRNAs cataloged in this database is low, as they are based on mRNA-seq data without RNase R enrichment, identified using CIRI2 and find_circ. Figure S2A-E illustrates the similarities and differences between circRNAs identified in our study and those recorded in ViruscircBase.

3. Relating to this and upon inspecting the provided code, I note that the authors use BWA mem in their alignments but perform separate alignments against the human and viral genomes. As BWA mem is a local aligner, this often leads to misalignments that can increase noise and artefact in the resulting data. The authors should instead perform a competitive alignment i.e. append the viral genome to the human genome FASTA and then align the sequencing data (as indeed the authors performed for the RNA-Seq analysis).

Response 2-3:

As advised by the reviewer, we have aligned the short-read and long-read circRNA sequencing data to reference genome, containing VZV and human genome, except for vircircRNA algorithm that was designed only for viral circRNA detection.

4. The specific regions enriched for putative VZV circRNAs (line 11, page 5) are interesting but the accompanying figure lacks any detail on the genomic structure and coding content. While hard to estimate from the plot, the peak around 11k might coincide with the R1 repeat region while the peaks at the right-hand end of the genome look to co-localize with the duplicated IRS and TRS repeats. Can the authors show these peaks are not simply artefacts aligning occurring from alignment to the repeat regions? In other words, did they allow for reads to map to multiple locations and if so, did they filter to only retain a single alignment per read?

Response 2-4:

To minimize the artifacts, we have added sequencing results for RNase R-digested and untreated samples in the revised manuscript (Figure 2F).

In the Herpesviridae family, there are commonly long repeat sequences exceeding 10k, posing a significant challenge for quantitative RNA expression. However, in VZV, the genomic DNA sequence directions of IRS and TRS are opposite, generating RNA sequences in different chain directions. For example, the genomic DNA sequence directions of ORF62 and ORF71 are completely opposite, but their transcribed ORF62 RNA is negative strand, and ORF71 RNA is positive strand. The short-read circRNA sequencing used in this study is strand-specific paired-end sequencing, and circRNA identification algorithms CIRI2, find_circ, and vircircRNA can accurately identify the chain direction of circRNA. Therefore, there should be no false positives in the circRNAs identified in the IRS and TRS regions due to these long oppositely oriented repeat sequences.

We observed that the region near the long unique regions ORF9 and ORF61 has a very high genomic coverage in RNase R-digested samples, consistent with the generation of a large number of circRNAs in the region near ORF61.

5. The genome-wide mutagenesis library would appear to be missing important spliced RNAs such as VLT and VLT63 which is a significant oversight. Moreover, it would be important to determine whether VZV circRNA biogenesis is significantly impacted by VLT/VLT63 deletion (or replacement with cDNA copies).

Response 2-5:

We thank the reviewer for valuable suggestions.

The design of circVLTslytic avoids ORFs, replacing only 20 base pairs upstream or downstream of the VLTs fifth exon. We have supplemented whole-genome sequencing, confirming that mutations only occur in the upstream and downstream regions of the fifth exon of VLTs in the entire VZV genome. Please refer to **Response 1-2**.

For viral RNA splicing, ViReMa analysis revealed that the majority of RNA splicing in pOka-M1 and pOka-M2 is the same as in pOka-WT (Figure S4G).

In addition, we confirmed that the mutations in pOka-M1 and pOka-M2 do not affect the splicing of VLTs and VLT63, VLTs splicing to ORF63 (Figure 7H-I).

6. The authors should delete the line suggesting that VLTs-associated circRNAs may play a regulatory role in latency as there is absolutely no data to show these are even made in latently-infected neurons, let alone serve a function (line 6, page 6)

Response 2-6:

It has been removed.

7. I'm afraid I'm rather lost on how the systematic experimental genome-wide scanning (line 10, page 7) was performed, let alone how this gave rise to 185 VZV circRNAs. The authors need to elaborate on this in both the results and methods sections.

Response 2-7:

Thank you for pointing out this issue. In order to better explain how we obtained experimental validation for the 200 VZV circRNAs, we have added a schematic diagram of the experimental

process in Figure 4A. Additionally, we have strengthened the description in the Results and Methods sections.

8. The authors should delete the line suggesting that VZV circRNAs have anything to do with VZV reactivation caused diseases, again due to a lack of any data whatsoever supporting this.

Response 2-8:

Thank you for pointing out this issue. We agree with the reviewer's suggestion and have removed it.

9. The authors are incorrect to state that pVLTs suppresses expression of ORF61 as the original study demonstrated this was mediated at the VLTs RNA (and not protein) level (line 4, page 12). Moreover, pVLTs is not detectable during latent infection and the studies relating to ORF61 expression were carried out using expression vectors that would not encode VLTs circRNAs, making this whole line of speculation incorrect.

Response 2-9:

We thank you the reviewer's careful reading of our MS. We corrected this error in the discussion section.

Minor concerns

1. The authors align against the wildtype pOka genome but are using a mutant pOka strain containing GFP and luciferase. Ideally, all alignments should be against this specific genome to reduce mismapping issues. It is also not described in the manuscript where the luciferase insertions. Ideally, a genome sequence for this mutant would be available and deposited at NCBI.

Response 2-10:

We thank the reviewer for pointing out this important issue. As the reviewer's suggestion, we performed a whole-genome sequencing to assemble the complete genome of our pOka strain containing GFP and luciferase. By using this fitting reference genome, we were able to identify additional VZV circRNAs. Now the whole-genome sequencing data has been deposited in NCBI SRA under the project accession number PRJNA1056528 (<https://www.ncbi.nlm.nih.gov/sra/PRJNA1056528>) and the assembled genomic sequences was accessible on NCBI GenBank under accession number PP054841 (<https://www.ncbi.nlm.nih.gov/nuccore/PP054841>). In addition, we also provided conversion for localization of VZV circRNAs in pOka (AB097933.1) and vOka (KU926314.1) strain (Data S1).

2. I would argue against the use of the term circVLTs as this could be interpreted as circRNAs that are expressed during viral latency (for which there exists no evidence to date).

Response 2-11:

We agree with the reviewer and used "circVLTs_{lytic}" instead.

3. The authors mentioned pooling biological replicates but I did not see any evidence provided that their biological replicates were sufficiently similar to do so.

Response 2-12:

We have added Figure 1B and Figure S2B-H to illustrate the identification of human and VZV circRNAs in each biological replicate sample. "pool_all" refers to the bioinformatics analysis of merged sequencing data after RNase R digestion for the identification of human and VZV circRNAs. Due to the generally low expression abundance of circRNAs, merging all samples (increasing sequencing depth) allows the identification of more VZV circRNAs. Figure S2B further illustrates that merging all samples reveals additional circRNAs that were not identified in individual samples.

4. Did the authors confirm the mutations (and absence of off-target mutations) in pOka-M1 and pOka-M2 by whole genome sequencing? This would be important to understand the differing results between M1 and M2.

Response 2-13:

We appreciate the reviewer's suggestion, and we have supplemented the whole-genome sequencing data to confirm the mutation of pOka-M1, and pOka-M2 (the data has been uploaded to NCBI SRA, <https://www.ncbi.nlm.nih.gov/sra/PRJNA1056528>). Assembled data indicated that none absence of off-target mutations.

Reviewer #3:

In this manuscript, "Identification and Functional Characterization of Varicella Zoster Virus Circular RNA in Lytic Infection," the authors performed sequencing on libraries made from RNase R treated RNA samples of VZV-infected materials. VZV-derived circular RNAs were identified through computational circRNA prediction algorithms, and VZV latency-associated transcripts (VLTs)-like circRNAs were targeted for experimental validation and functional studies. For the latter a BAC system was used to disrupt the expression of the circVLTs by mutating sequences flanking the circVLTs's splice donor site. It was concluded that disruption of the circVLTs did not affect virus replication, but lead to increased acyclovir sensitivity in in vitro culture of human skin tissues and a SCID mouse xenograft model. Generally, the computational identification of VZV circular RNAs is an advance in the field by defining a new group of RNA elaborated by VZV. However, the experimental studies of the VZV circular RNAs do not sufficiently support the authors conclusions.

Response 3-1:

We appreciate the reviewer for the positive comments. We have performed several important experiments to address the reviewer's concerns.

With respect to experimental confirmation of the identity of VZV circRNAs. Several things are requested:

1. *Figure 2A: This figure only shows genome coverage with RNase R digested RNA samples from VZV-infected materials. VZV coverage with RNA not treated with RNase R should also be included/overlaid for comparison to rule out any RNA sequence bias. The peaks identified should be scaled by ORF boundaries*

Response 3-2:

For consistency, we have re-supplemented the sequencing results with or without RNase R treatment and added annotations for ORF boundaries (Figure 2A).

2. *Figure 3A: It would be helpful if peaks identified in Fig. 2A can be related to the putative circular RNAs identified. Do they peaks and circRNAs overlap significantly—do they correspond meaningfully. How are the reconstructions of the "Full" circular RNAs obtained? Is this by divergent primer PCR followed by sequencing. It is critical to state whether this is done by alignment of reads from initial sequencing or by subsequent experimental confirmation.*

Response 3-3:

Acknowledging the reviewer's suggestion, in the vicinity of ORF61 (110000-115000), significant enrichment in genome coverage peaks was observed after RNase R digestion, indicating abundant VZV circRNA. We have incorporated this description into the results.

Reconstructions of the "Full" circular RNAs were performed by the CIRI-full software (<https://ciri-cookbook.readthedocs.io/en/latest/CIRI-full.html#running-ciri-full-pipeline>) for the full-length of VZV circRNA from short-read sequencing data. We have added descriptions in the methods and results. To enhance the accuracy of VZV circRNA full-length reconstruction, we supplemented with RNase R digestion, followed by rolling circle reverse transcription and Nanopore long sequencing (Figure 2B and Figure 3A).

Additionally, as shown in Figure 4C-D, we designed divergent primers at exon 1 (VLTs exon 2) of circVLTs_{lytic}, obtained the full-length sequence of circVLTs_{lytic} through TA cloning and Sanger sequencing (Figure 4E).

3. Figure 4C: The labeling of the bands (red arrowheads and number) completely obliterates whether weak bands are seen in mock lanes at corresponding positions. P3 looks not viral, since band appears in mock lane – similar comment on P8. Were these PCR products sequenced? Since the authors are concentrating in circVLTs and full map of isoforms that are sequence confirmed is needed.

Response 3-4:

We have made adjustments to Figure 4, which now includes validation of short- and long-read circRNA sequencing identification results (Figure 4B and Figure S3). We have added verification of the full length of circVLTs_{lytic} (Figure 4C-D), and Figure 4E displays the results of Sanger sequencing for circVLTs_{lytic} full length.

4. Figure 6: Bands identified as circRNA from the 6 different patients need qPCR done with RNase R treated and untreated conditions, and sequenced to confirm they are predicted backspliced amplicons.

Response 3-5:

Typically, a sufficient amount of RNA, approximately 4 µg total RNA is required to complete RNase R digestion and purification. Due to the limited vesicle fluid isolated from herpes zoster patients that was challenging to extract DNA for confirming VZV infection and RNA for RNase R digestion. Instead, we isolated three clinical strains of VZV from the skin of patients with acute herpes zoster. We extracted RNA from these VZV clinical strains infected-ARPE-19 cells and performed RNase R digestion. Inverse RT-PCR and Sanger sequencing results indicated that the expressed circRNA was consistent with the pOka strain.

With respect to the assertion that disruption of circVLTs confers increased acyclovir sensitivity. Fussing with any viral sequence requires detailed examination of "off-target" effects.

1. Does the mutation introduced into the BAC include any part of the canonical splice sequence?

Response 3-6:

We supplemented the whole-genome sequencing data for pOka-WT, pOka-M1, and pOka-M2, and assembled their genomic sequences (the data has been uploaded to NCBI SRA,

<https://www.ncbi.nlm.nih.gov/sra/PRJNA1056528>), confirming that BAC recombination only affected the splicing site of the fifth exon of circVLTs_{lytic}, specifically 20 nucleotides upstream or downstream of position 114003.

2. The changes in circVLTs isoforms as well as linear transcript need to be enumerated and documented.

Response 3-7:

We would like to say thank you for the valuable suggestion. The characterize of circVLTs_{lytic} (Figure 6D-E and Data S4), VLTs and VLT63 (Figure H-I) in the mutations of pOka-M1 and pOka-M2 was added. Please see additional comments to **Response 1-2** and **Response 2-5** for details.

3. Does introduction in trans of the linear VLTs reconstitute wild-type phenotype?

Response 3-8:

This is a great question.

The CircVLTs_{lytic} contains more than 10 isoforms sharing the same back-splice junctions (BSJs). In this study, we investigated the functions of these isoforms by mutating the 5' terminal 20bp sequence of CircVLTs_{lytic}, affecting their expression. However, it is currently unclear which specific isoform or combination of isoforms plays a crucial role. Answering this question may require overexpression or restoring mutations in individual isoforms or different combinations. We plan to undertake this intricate task in future research.

REVIEWER COMMENTS

Reviewer #1 (Remarks to the Author):

Using RNase R and long read sequencing improves the validity of this revised manuscript.

The genomic traces in Fig. 2F with the various conditions are informative and support of their conclusions.

Distracting typos remain of "find_cric" in some figures and "find_circ" in other figures. See Fig. 1A, B for examples.

This reviewer asked for quantitative methods to measure levels of circular RNAs with real-time PCR or digital PCR, but the authors did not provide these results in the revised document.

Reviewer #2 (Remarks to the Author):

The authors have made significant efforts to address the comments and critiques provided by the reviewers and the resulting manuscript has been significantly strengthened and the findings remain (potentially) very exciting.

However, I am still concerned regarding point 4 (response 2-4) from the original review. The amended version of Figure 2F is still extremely hard to interpret as it aims to show coverage plots for 4 short-read datasets (3 with RNase R treatment) and one long-read dataset (also with RNase R treatment). However it is not possible to examine the profiles of the individual datasets due to them being all overlaid and the scaling not taking into account the different number of reads in each library. What is needed to resolve this is to plot each track separately in a strand separated manner (potentially as a supplementary figure) with the coverage scaled appropriately for each. My driving concern here is that the 72h + RNase R data shows an overall profile that is not unlike those I have observed for standard VZV RNASeq analyses and it is critical to be able to directly compare the 72h datapoints +/- RNase R treatment. It is also curious that the profiles for 24h and 48h + RNase look relatively similar (Fig 2A in original manuscript) but so different from 72h + RNase R. It would be interesting to know whether the human circRNAs observed in the 72h + RNase R sample were similar to 24h and 72h.

The second point in this regard is that the VZV genome encode multiple short reiterations regions (R1, R2, R3, R4, R5) that can produce significantly misalignments. It is thus essential to include these loci on the genome map (Fig. 2F) and to determine whether these loci localize with abundant circRNAs. From the existing plot it is hard to be sure but the location of the R1 region (~13-13.5kb) appears close (potentially overlapping) with the large peak between 10-15kb.

I am also curious about the 72h - RNase R dataset as it appears there are two dominant peaks either side of the BAC vector region. This is somewhat unusual and looks like a potentially alignment artefact. The authors should confirm that the same VZV strain was used in the experiments and the same pipeline used in the analysis.

Reviewer #3 (Remarks to the Author):

The authors' attempt to address all three reviewers' comments are admirable. Unfortunately, for this reviewer's comments, the authors are not able to satisfactorily respond with experimental validations asked for. Reconstitution and complementation experiments are kicked down the road for future experimentation. And a lack of sufficient patient materials are cited as reason for precluding definitive studies. It has not been validated that this viral genome produces much more circRNAs than previously thought. Finally, the functional characterization of VZV circRNA stated in the title is not substantiated.

Reviewer #1:

Using RNase R and long read sequencing improves the validity of this revised manuscript. The genomic traces in Fig. 2F with the various conditions are informative and support of their conclusions.

Response 1-1: We thank the reviewer for the positive comments. Please be noted that the previous Figure 2F has been moved to Supplementary Figure 3 for more clearly examine the profiles as suggested by the reviewer 2.

Distracting typos remain of "find_cric" in some figures and "find_circ" in other figures. See Fig. 1A, B for examples.

Response 1-2: We have corrected this error in all the figures. We thank the reviewer for the careful review.

This reviewer asked for quantitative methods to measure levels of circular RNAs with real-time PCR or digital PCR, but the authors did not provide these results in the revised document.

Response 2-3: We performed the real-time PCR (qPCR) to measure levels of VZV circular RNAs, and the results are shown in Supplemental Figure 5F-G.

Reviewer #2:

The authors have made significant efforts to address the comments and critiques provided by the reviewers and the resulting manuscript has been significantly strengthened and the findings remain (potentially) very exciting.

Response 2-1: We appreciate the reviewer for the positive comments.

However, I am still concerned regarding point 4 (response 2-4) from the original review. The amended version of Figure 2F is still extremely hard to interpret as it aims to show coverage plots for 4 short-read datasets (3 with RNase R treatment) and one long-read dataset (also with RNase R treatment). However it is not possible to examine the profiles of the individual datasets due to them being all overlaid and the scaling not taking into account the different number of reads in each library. What is needed to resolve this is to plot each track separately in a strand separated manner (potentially as a supplementary figure) with the coverage scaled appropriately for each.

Response 2-2: We agree with the reviewer and the reviewer's feedback and valuable suggestions. In response, we have relocated Figure 2F to Supplementary Figure 3 and have organized it separately with each dataset.

My driving concern here is that the 72h + RNase R data shows an overall profile that is not unlike those I have observed for standard VZV RNASeq analyses and it is critical to be able to directly compare the 72h datapoints +/- RNase R treatment. It is also curious that the profiles for 24h and 48h + RNase look relatively similar (Fig 2A in original manuscript) but so different from 72h + RNase R. It would be interesting to know whether the human circRNAs observed in the 72h + RNase R sample were similar to 24h and 48h.

Response 2-3:

We agree with the reviewer. The genomic coverage of the 72h + RNase R data shows significant variability, with nearly all genes exhibiting high transcription levels, particularly in the vicinity of ORF9 and ORF9A, where RNA abundance is exceptionally high (Supplementary Figure 3). To further validate the results of the 72h datapoints with and without RNase R treatment, we conducted additional short-read sequencing in this revised manuscript. The results are consistent with our previous findings (Supplementary Figure 1D).

Cytopathic effects in SH-SY5Y cells infected with VZV at 24h and 48h were observed to be 30% and 50%, respectively (Supplementary Figure 1A). However, at 72h post-infection, cytopathic effects approached 90%, coinciding with a higher abundance of viral RNA within the cells (Supplementary Figure 1D and Supplementary Figure 3D). However, at this stage, cells were nearing death, and their state likely underwent significant changes. This also explains the minimal number of human circRNAs identified in the 72h + RNase R group (as shown in the figure below). Hence, we chose to analyze host cell circRNAs induced by VZV infection during the early stages (24h, 48h) of infection.

The second point in this regard is that the VZV genome encode multiple short reiterations regions (R1, R2, R3, R4, R5) that can produce significantly misalignments. It is thus essential to include these loci on the genome map (Fig. 2F) and to determine whether these loci localize with abundant circRNAs. From the existing plot it is hard to be sure but the location of the R1 region (~13-13.5kb) appears close (potentially overlapping) with the large peak between 10-15kb.

Response 2-4: We thank the reviewer for bringing this to our attention. In Figure 3A, Supplementary Figure 3 and Supplementary Figure 6G, we included both multiple short reiterations regions (R1, R2, R3, R4, R5) proposed by Jensen NJ [Virology, 2020, PMID: 32452416] and the unique and repeat regions (TRL, IRL, UL, IRS, US, TRS) in loci on the VZV genome map. Our results did not support a correlation between the biogenesis of VZV circRNAs and the multiple short reiteration regions.

I am also curious about the 72h – RNAse R dataset as it appears there are two dominant peaks either side of the BAC vector region. This is somewhat unusual and looks like a potentially alignment artefact. The authors should confirm that the same VZV strain was used in the experiments and the same pipeline used in the analysis.

Response 2-5: We appreciate the reviewer's thorough evaluation. We resequenced the libraries of VZV infected with 72h+RNAse R and 72h - RNAse R, and the results were consistent with previous studies (Supplementary Figure 1D). The BAC vector region contains a GFP gene driven by CMV promoter, which can transcribe RNA and express GFP protein in infected cells (Supplementary Figure 1A). Another possibility, in our

opinion, is that the BAC vector DNA sequences may undergo transcription in mammalian cells. This aspect will be scrutinized in our future work.

Reviewer #3:

The authors' attempt to address all three reviewers' comments are admirable. Unfortunately, for this reviewer's comments, the authors are not able to satisfactorily respond with experimental validations asked for. Reconstitution and complementation experiments are kicked down the road for future experimentation.

Response 3-1: We appreciate both the previous and current suggestions from the reviewer and apologize for not addressing them in the previous version.

To address the reviewer's request regarding the introduction of in trans circVLT reconstitute of the wild-type phenotype of pOka-M1 to further confirm the function of circVLT, we overexpressed circVLT 419 nt or in circVLT 542 nt isoforms in SH-SY5Y cells using a lentiviral circular RNA overexpression vector. Subsequently, the cells were infected with pOka-M1 (Supplementary Figure 7A-D). Our results showed that the overexpression of either isoform (circVLT 419 nt or in circVLT 542 nt) promoted the replication of pOka-M1 and alleviated the antiviral effect of ACV.

And a lack of sufficient patient materials are cited as reason for precluding definitive studies.

Response 3-2: We appreciate the critical suggestion from the reviewer. However, obtaining sufficient RNA from the patient's herpes fluid for RNaseR digestion is indeed currently challenging. To better approximate the clinical study with sufficient RNA, alternatively, we isolated VZV clinical strains for verification of VZV-encoded circRNA, both with and without RNase R treatment. Results from inverse RT-PCR and Sanger sequencing indicated the presence of VZV-encoded circRNAs in these clinical strains (Figure 5E). Two VZV clinical strains have been assembled using whole genome sequencing, and the genome sequence information have been uploaded to Genbank under accession numbers PP261331 and PP261332.

It has not been validated that this viral genome produces much more circRNAs than previously thought.

Response 3-3:

Thank you for your continued feedback. We acknowledge the reviewer's comments. Initially, we did not anticipate VZV to generate a significantly higher number of circRNAs. We believe that VZV encodes a comparable number of circRNAs to other viruses, such as human cytomegalovirus and various coronaviruses. In our previous work, we discovered 351, 224, and 2764 viral circRNAs in cells infected with SARS-CoV-2, SARS-CoV, and MERS-CoV, respectively (Yang S, 2022, J Med Virol, PMID: 35318674). In another alpha-herpesvirus, HCMV, we identified 324 viral circRNAs through TA cloning (Yang S, 2022, Microbiol Spectr, PMID: 35604147). In this experiment, we enriched circRNA using RNase R followed by deep short-reads sequencing and long-reads sequencing. Bioinformatic analyses identified 106 (CIRI2), 2806 (find_circ), 305 (vircircRNA), and 1358 (CIRI-long) viral circRNAs, which were further confirmed by TA cloning through PCR, resulting in the identification of 200 viral circRNAs. This suggests that the production of a large number of circRNAs by viruses may be a common phenomenon.

Finally, the functional characterization of VZV circRNA stated in the title is not substantiated.

Response 3-4: We thank the reviewer for the helpful comments. We have modified the conclusions related to the functional role of circRNAs and amend the title accordingly.

REVIEWERS' COMMENTS

Reviewer #1 (Remarks to the Author):

The authors now include qPCR results, as requested, but their results in Supplementary File 5F raise concerns about the specificity of the primers. For example, in the mock conditions they report signal from multiple divergent primer pairs for viral circular RNAs in the mock (uninfected) conditions. There should be no viral transcripts in mock conditions, which raises the possibility that these primers are amplifying transcripts that are not viral circular RNAs.

Reviewer #2 (Remarks to the Author):

The authors have provided satisfactory rebuttals to my question. This remains a fascinating piece of work that raises many interesting questions about VZV biology that I hope the authors will continue to pursue.

Reviewer #1:

The authors now include qPCR results, as requested, but their results in Supplementary File 5F raise concerns about the specificity of the primers. For example, in the mock conditions they report signal from multiple divergent primer pairs for viral circular RNAs in the mock (uninfected) conditions. There should be no viral transcripts in mock conditions, which raises the possibility that these primers are amplifying transcripts that are not viral circular RNAs.

Response 1: We acknowledge the reviewer's valid concerns regarding the specificity of the primers. In response, we have taken steps to improve their specificity by redesigning eight pairs of qPCR primers for VZV circRNAs. These newly designed primers yield products ranging from 100bp to 200bp, with increased length and temperature melting (TM) of primers. The results, as depicted in **Supplementary Figure 5F-G**, indicate a significant enhancement in specificity.

Reviewer #2:

The authors have provided satisfactory rebuttals to my question. This remains a fascinating piece of work that raises many interesting questions about VZV biology that I hope the authors will continue to pursue.

Response 2: We thank the reviewer for the meticulous evaluation of our manuscript and the highly constructive suggestions, which undeniably enhanced its quality. We acknowledge that this study raises numerous intriguing questions, and we are committed to further exploring the biological functions of VZV circRNAs in our ongoing research efforts.